# On Bilingual Lexicon Induction with Large Language Models

**Yaoyiran Li    Anna Korhonen    Ivan Vulić**
Language Technology Lab, TAL, University of Cambridge
{yl711,alk23,iv250}@cam.ac.uk

## Abstract

Bilingual Lexicon Induction (BLI) is a core task in multilingual NLP that still, to a large extent, relies on calculating cross-lingual word representations. Inspired by the global paradigm shift in NLP towards Large Language Models (LLMs), we examine the potential of the latest generation of LLMs for the development of bilingual lexicons. We ask the following research question: *Is it possible to prompt and fine-tune multilingual LLMs (mLLMs) for BLI, and how does this approach compare against and complement current BLI approaches?* To this end, we systematically study **1)** zero-shot prompting for unsupervised BLI and **2)** few-shot in-context prompting with a set of seed translation pairs, both without any LLM fine-tuning, as well as **3)** standard BLI-oriented fine-tuning of smaller LLMs. We experiment with 18 open-source text-to-text mLLMs of different sizes (from 0.3B to 13B parameters) on two standard BLI benchmarks covering a range of typologically diverse languages. Our work is the first to demonstrate strong BLI capabilities of text-to-text mLLMs. The results reveal that few-shot prompting with in-context examples from nearest neighbours achieves the best performance, establishing new state-of-the-art BLI scores for many language pairs. We also conduct a series of in-depth analyses and ablation studies, providing more insights on BLI with (m)LLMs, also along with their limitations.

## 1   Introduction and Motivation

Bilingual Lexicon Induction (BLI), also known as word translation, is a fundamental research topic in multilingual NLP that aims to bridge the lexical gap between languages (Ruder et al., 2019). It has a wide range of applications such as machine translation (Artetxe et al., 2018b; Marchisio et al., 2020; Chronopoulou et al., 2021) and cross-lingual transfer learning, especially for low-resource languages (Sun et al., 2021; Zhou et al., 2021; Wang et al., 2022). Over the past decade, state-of-the-art

(SotA) BLI approaches have been predominantly supported by learning a cross-lingual word embedding (CLWE) space, with which BLI is tackled via nearest neighbour retrieval (Artetxe et al., 2018a; Heyman et al., 2019; Peng et al., 2021; Li et al., 2022a; Marchisio et al., 2022, *inter alia*).

Meanwhile, autoregressive text-to-text large language models (LLMs) have emerged as the cornerstone of cutting-edge NLP research (Raffel et al., 2020; Brown et al., 2020; Ouyang et al., 2022; Chowdhery et al., 2022). For example, multilingual LLMs (mLLMs) have shown (sentence-level) machine translation capabilities (Vilar et al., 2022; Briakou et al., 2023), although they have not been pretrained for machine translation in a supervised manner. Motivated by the recent remarkable success of (m)LLMs, in this work we investigate **1)** the potential of prompting and fine-tuning of mLLMs for BLI and **2)** how their capabilities compare against and complement current BLI approaches. We focus on how to expose word-level bilingual knowledge and elicit word translations from multilingual LLMs. To our best knowledge, we are the *first* to leverage autoregressive mLLMs for BLI.[1]

We systematically study zero-shot and few-shot prompting for BLI with off-the-shelf encoder-decoder and decoder-only autoregressive mLLMs (Radford et al., 2019; Raffel et al., 2020; Brown et al., 2020), respectively. In the few-shot scenario, we propose to incorporate in-context examples from nearest neighbours into the prompts to boost the BLI performance. In order to guide the mLLMs' generation, we hand-craft *'mask-filling-style'* and *'GPT-style'* templates catering to the characteristics of different LLMs and conduct extensive template search for BLI. In addition to pro-

---

[1]We point out that the work of Li et al. (2022a) leverages only the encoder part of mT5 (the decoder is dropped) in one of their experiments to extract CLWEs for BLI. In contrast, our experiments with mT5 utilise its full encoder-decoder structure and generate the target words autoregressively, completely different from CLWE-based approaches.

viding a complete and effective pipeline for BLI via prompting off-the-shelf mLLMs, we also investigate BLI-oriented fine-tuning with the LLMs' own pretraining objectives, aiming at specialising mLLMs into 'few-shot word translators'.

We conduct extensive experiments on two standard BLI benchmarks, XLING (Glavaš et al., 2019) and PanLex-BLI (Vulić et al., 2019), investigating the word translation capabilities of off-the-shelf mLLMs (we adopt 18 models from 5 LLM families) in various BLI setups. Our comprehensive comparisons between mLLMs confirm, as expected, that **1)** different LLM families display varying word translation capabilities and **2)** stronger BLI performance tends to be associated with larger model sizes. To demonstrate the effectiveness of our prompt-based approach, we benchmark our method against two SotA CLWE-based baselines. Notably, our approach with LLaMA$_{13B}$ outperforms the CLWE-based SotA on the XLING dataset by a considerable margin, establishing new SotA results on many language pairs in all BLI setups. Meanwhile, we also identify two limitations of BLI with mLLMs: **1)** they are less competitive on the PanLex-BLI benchmark for lower-resource languages; **2)** CLWE-based approaches usually support more languages than mLLMs. Finally, we run a series of insightful ablations and discuss the usefulness of BLI-oriented fine-tuning. In short, our work validates the BLI capabilities of mLLMs and proposes new methodology for BLI. We hope that the combination of our comprehensive analyses and discussions, including on limitations, will pave the way for the development of stronger BLI systems in the future. Our code is publicly available at `github.com/cambridgeltl/prompt4bli`.

## 2 Related Work

**Bilingual Lexicon Induction.** Over the past decade, predominant BLI approaches have relied on the calculation of cross-lingual word embeddings (CLWEs) where, in the most popular BLI variant, two transformation functions are learned to respectively map source and target monolingual static word embedding spaces into a shared cross-lingual space (Xing et al., 2015; Lample et al., 2018; Joulin et al., 2018; Artetxe et al., 2018a; Alvarez-Melis and Jaakkola, 2018; Patra et al., 2019; Mohiuddin et al., 2020; Glavaš and Vulić, 2020; Peng et al., 2021; Li et al., 2022a; Marchisio et al., 2022). Then, relying on the learned CLWE

space, BLI has been conducted via nearest neighbour retrieval. A detailed overview of different BLI principles can be found, e.g., in the work of Ruder et al. (2019).

More recently, researchers have attempted BLI by leveraging *encoder-only* multilingual masked language models (mMLMs) such as mBERT (Devlin et al., 2019) and XLM-R (Conneau et al., 2020) whose neural architecture consists of only Transformer encoders (Vaswani et al., 2017). Gonen et al. (2020) prompt mBERT with templates where the target word is replaced with a '<mask>' token, and the language modelling head of mBERT outputs a subword token to fill the mask. This method is theoretically flawed because it cannot address the cases where the target word comprises two or more subword tokens. Therefore, Gonen et al. (2020) only evaluate BLI on a small set of 'toy' examples rather than standard BLI datasets. In terms of performance, this method lags far behind traditional BLI approaches. A more successful way of leveraging mMLMs is to extract decontextualised word representations from them (Zhang et al., 2021). The strongest CLWEs for BLI so far are learned via a two-stage contrastive approach combining both static (e.g., fastText) and mMLM-extracted features (Li et al., 2022a).[2]

**Text-to-Text LLMs.** Autoregressive LLMs have established new state-of-the-art results on many NLP tasks. The prominent model groups include **1)** *encoder-decoder* LLMs such as BART (Lewis et al., 2020) and T5 (Raffel et al., 2020); **2)** OpenAI's *decoder-only* GPT series such as GPT-2 (Radford et al., 2019), GPT-3 (Brown et al., 2020), and InstructGPT (Ouyang et al., 2022); **3)** other GPT-like LLMs with specific improvements such as Chinchilla (Hoffmann et al., 2022), PaLM (Chowdhery et al., 2022), and LLaMA LLM series (Touvron et al., 2023).

Our work adopts five families of open-source text-to-text multilingual LLMs for BLI, including mT5 (Xue et al., 2021), mT0 (Muennighoff et al., 2022), XGLM (Lin et al., 2022), mGPT (Shliazhko et al., 2022), and LLaMA (Touvron et al., 2023). We introduce each of these in more detail in §3.1. Unlike the encoder-only MLMs, text-to-text LLMs are theoretically capable of generating words consisting of arbitrary numbers of subword tokens.

---

[2]Using mMLMs alone still underperforms purely fastText-based methods since mMLMs are contextualised encoders pretrained for sentence-level tasks (Li et al., 2022a).

## 3 Methodology

**BLI Task: Preliminaries and Terminology.** Assuming a bilingual scenario with a source language $L^x$ and a target language $L^y$ with their respective vocabularies denoted as $\mathcal{X}$ and $\mathcal{Y}$, the BLI task is typically formulated as a standard information retrieval task (Gaussier et al., 2004; Glavaš et al., 2019). The goal is to rank the words from $\mathcal{Y}$ with respect to their similarity to the input source word $w^x$. The vocabulary size for each language is typically set to 200k (Li et al., 2022a), covering the most frequent 200k word types in each language. A bilingual lexicon then comprises a set of one-to-one source and target word translation pairs (Mikolov et al., 2013), and we denote a word pair as $\pi = (w^x, w^y)$ where $w^x \in \mathcal{X}, w^y \in \mathcal{Y}$.

We assume a set $\mathcal{D}_S$ of $N$ available seed translation pairs, constituting the so-called *seed dictionary*, which are used as the training set. Depending on the number of training pairs, the task is usually referred to as *supervised BLI* (typically, $N \geq 5K$), *semi-supervised BLI* (e.g., $0 < N \leq 1K$), and *unsupervised BLI* ($N = 0$) in the literature (Artetxe et al., 2018a; Zhao et al., 2020; Li et al., 2022a). For convenience, we also refer to the unsupervised setup as *zero-shot BLI* ($N = 0$) and denote the setup with a handful of seed translation pairs as *few-shot BLI* ($N > 0$), corresponding to how we prompt mLLMs for BLI (we describe zero-shot and few-shot prompts for BLI later in §3). A test set $\mathcal{D}_T$, where $\mathcal{D}_S \cap \mathcal{D}_T = \emptyset$, is used for evaluation.

In some cases, a source word may have more than one ground-truth translation (i.e., there exist two or more word pairs in a BLI dictionary that share the same source word). Following previous work (Lample et al., 2018; Glavaš et al., 2019; Li et al., 2022a), we consider a prediction correct as long as it is any of the ground-truth translations. The BLI scores are reported based on the standard Precision@K (P@K) BLI measure, where $K$ denotes the length of the ranked list.

### 3.1 Prompting Multilingual LLMs for BLI

This study employs five families of mainstream multilingual text-to-text LLMs (mLLMs): mT5, mT0, XGLM, mGPT, and LLaMA.[3] Based on their model structures, we group these models into two

categories; in what follows, we briefly introduce each of them and showcase some simple templates used for *'BLI-prompting'* the LLMs.

The first category includes mT5 and mT0, two *encoder-decoder* LLM families that leverage the full Transformer architecture (Vaswani et al., 2017). Each model family comes in five different sizes, and we evaluate all these ten models.

- **mT5** (Xue et al., 2021) is pretrained on the mC4 dataset covering 101 languages. The LLM leverages a *span-corruption* objective that tries to reconstruct consecutive spans of dropped-out tokens replaced with special mask tokens.

- **mT0** (Muennighoff et al., 2022) is a multitask-finetuned mLLM based on instruction fine-tuning from the original mT5 model. The fine-tuning is conducted with English prompts on mT0's xP3 dataset spanning 46 languages.[4]

For these two encoder-decoder style mLLMs, we aim to derive prompts such that the first word of the output sequence serves as its guess for $w^y$. Catering to its span-corruption objective, for mT5 we propose to design *mask-filling-style* English templates where '<mask>' tokens are used as placeholders for the target words. Here is an example template: *'The $L^x$ word $w^x$ in $L^y$ is <mask>.'*, where $L^x$, $L^y$, and $w^x$ are placeholders for the source language, target language, and the input source word, respectively.[5] When a prompt based on this template is fed into mT5, its decoder will then output a sequence to fill the mask. Since mT0 is based on mT5, we found that mask-filling-style prompts are also applicable to mT0. However, unlike for mT5, the instruction-tuned mT0 fits templates without the '<mask>' token.[6] For simplicity, we will denote all such templates without any '<mask>' tokens as *'GPT-style templates'*.

The second model category comprises XGLM, mGPT, and LLaMA as three *decoder-only* LLMs pretrained with causal LM losses. Our experiments involve five XGLM and two LLaMA models whose

---

[3]We also experimented with mBART (Liu et al., 2020). However, we do not report the results with mBART since its BLI performance proved inferior in our preliminary investigations.

[4]Among all the LLMs covered in our work, mT0 is the only one trained for sentence-level translation tasks (machine translation is one of the tasks during its multitask fine-tuning). However, our experimental results reported later indicate that this does not benefit BLI in our prompting setups.

[5]The '<mask>' token for mT5 and mT0 is actually '<extra_id_0>'. Therefore, an example of an actual prompt would be *'The German word gebouw in French is <extra_id_0>.'*.

[6]For instance, *'The $L^x$ word $w^x$ in $L^y$ is'* may prompt mT0 to output $w^y$ to complete the input sentence, and *'How do you say $w^x$ in $L^y$?'* would prompt mT0 to generate $w^y$ to answer the question.

model sizes are no larger than 13B parameters, while mGPT only releases one model of size 1.4B. Unlike encoder-decoder LLMs for conditional generation, the decoder-only causal LLMs first repeat the input sequence in their output, and we construct prompts that induce LLMs to produce $w^y$ immediately after the repeated input sequence.

• **XGLM** (Lin et al., 2022) offers multilingual LLMs similar to GPT-3 (Brown et al., 2020) and is reported to outperform GPT-3 of comparable size in a series of tasks. The work builds a CC100-XL dataset based on Conneau et al. (2020) and Wenzek et al. (2020), and XGLM is pretrained with a subset of it covering 30 languages.

• **mGPT** (Shliazhko et al., 2022) reproduces the GPT-3 structure and is trained on 60 languages using Wikipedia and C4 data (Raffel et al., 2020).

• **LLaMA** (Touvron et al., 2023) is a recently released SotA LLM family trained on trillions of tokens exclusively from publicly available datasets; it supports 20 languages. LLaMA also features its efficient implementation, and it adopts a series of recent improvements on normalisation, activation functions, and positional embeddings.

Our decoder-only LLMs solely leverage GPT-style prompts introduced above for mT0, since their tokenisers usually do not support '<mask>' tokens.

### 3.2 Retrieval-Augmented In-Context Learning

In §3.1, we presented some simple zero-shot prompts (i.e., prompts without in-context examples) for BLI. However, recent work highlights the few-shot capabilities of modern LLMs (Brown et al., 2020). Therefore, we also investigate few-shot templates for improved BLI performance.[7]

We propose to retrieve the nearest neighbours of a source word which we use to construct in-context samples to boost BLI performance. More specifically, given $\mathcal{D}_S$ and an input source word $w^x$, we extract $n$ word pairs $(w_i^x, w_i^y) \in \mathcal{D}_S, 1 \leq i \leq n$, such that $w_i^x, 1 \leq i \leq n$ are $n$ nearest neighbours of $w^x$ in the auxiliary static monolingual word embedding space of $\mathcal{X}$. This auxiliary space is based on pretrained fastText word embeddings (Bojanowski et al., 2017)[8] and we use the cosine

similarity measure for the retrieval.[9] We again design mask-filling-style and GPT-style few-shot templates for the mLLMs, as discussed in §3.1. Similar to zero-shot prompts, for few-shot prompts we also extract *the first word* after removing special tokens (e.g., start-of-sentence, padding, and '<mask>' tokens) and repeated input sequence (for decoder-only models) as the prediction of $w^y$.

### 3.3 Template Design and BLI Inference

**Template Design.** We hand-craft in total 102 English zero-shot and few-shot templates, respectively listed in Tables 10 and 11 of Appendix C. A small set of basic templates is fully manually designed, and additional variants are then created by modifying or replacing the punctuation (see the tables). For each LLM, we search for its best zero-shot template and best few-shot template on a randomly chosen language pair (German, French) and fix the template choices for experiments on all other language pairs. The best template choices for each LLM are provided in Table 12 (Appendix C).

**BLI Inference.** At inference, we adopt beam search for both encoder-decoder and decoder-only LLMs and make the generator return the final beam ranked by their sequence scores. For each input prompt corresponding to $w^x$, we iterate through the returned set of sequences, and for each sequence we extract the word after removing any redundant prefix content, as described in §3.2. The first word extracted that appears in the target vocabulary is returned as our prediction of $w^y$.

### 3.4 BLI-Oriented Fine-Tuning

This work predominantly focuses on 'learningless' experiments based on zero-shot and few-shot in-context setups with off-the-shelf mLLMs for BLI without any fine-tuning. As a side experiment, we also aim to fine-tune smaller-scale mLLMs, making them specialise into few-shot word translators with our few-shot prompts as input. Our training set is still $\mathcal{D}_S$, but we now exclude retrieving an input $w^x$ itself as an in-context example. We combine the $\mathcal{D}_S$ of each language pair with which we fine-tune encoder-decoder mLLMs with mT5's span-corruption loss and fine-tune decoder-only LLMs with the standard causal LM objective.

---

[7]Note again that few-shot in-context learning does not require any actual fine-tuning of LLMs. The word 'learning' here only refers to inserting in-context examples into the input prompt sequence.

[8]See Appendix B for more details about the fastText WEs used in our work.

---

[9]For the rare cases where an in-context source word retrieved may have more than one translation in $\mathcal{D}_S$, we only keep the target word with the highest word frequency in fastText's training data.

## 4 Experimental Setup

**Training and Evaluation Data.** Our experiments adopt two standard and publicly available BLI datasets, also used in a body of very recent BLI research (Vulić et al., 2020; Sachidananda et al., 2021; Aboagye et al., 2022; Li et al., 2022a,b; Vulić et al., 2023). **1)** XLING (Glavaš et al., 2019) provides BLI dictionaries covering 8 languages and 56 BLI directions. Among these 8 languages, 5 are supported by all of our mLLMs: English (EN), French (FR), German (DE), Italian (IT), and Russian (RU). Therefore, §5 mainly focuses on and reports results on all the $20 = 5 \times 4$ BLI directions for the 5 languages.[10] For each language pair, XLING provides a test set $\mathcal{D}_T$ of 2K translation pairs. It also provides training sets $\mathcal{D}_S$ of 5K and 1K translation pairs, where the former is the superset of the latter. For brevity, we denote the cases $|\mathcal{D}_S| = 5$K, $|\mathcal{D}_S| = 1$K, and $|\mathcal{D}_S| = 0$ as the 5K setup, 1K setup, and unsupervised setup, respectively.[11] **2)** PanLex-BLI (Vulić et al., 2019) offers BLI lexicons spanning 15 lower-resource languages and all 210 BLI directions. We select three languages that are supported by most of our mLLMs: Bulgarian (BG), Catalan (CA), and Hungarian (HU). The test set size of PanLex-BLI is also 2K; under the lower-resource assumption, we only focus on unsupervised and 1K BLI setups.

**Main Experiments.** In our main experiments, we prompt 18 off-the-shelf models from 5 mLLM families mentioned in §3.1[12] for BLI *without* any fine-tuning[13] and systematically evaluate their BLI performance in three different BLI setups on XLING and PanLex-BLI datasets introduced above. In 5K and 1K setups, 5-shot in-context learning is adopted for our mLLMs, while in the unsupervised setup, zero-shot prompts are used. We compare the BLI scores between different mLLMs from the perspectives of LLM family and model size, and we also benchmark their performance against two SotA CLWE-based baselines, introduced later. Selected results are summarised in §5.1 while full and detailed BLI scores are reported in Appendix D.

**Side Experiments.** We conduct a series of additional experiments to further understand the BLI capabilities of mLLMs. **1)** We investigate how the BLI performance is related to the number of in-context examples (5K and 1K setups). **2)** As an ablation study, we validate the usefulness of (our proposed) in-context samples extracted from nearest neighbours by comparing with randomly sampled in-context examples. **3)** Finally, we fine-tune some of our relatively smaller-scale LLMs, including mT5$_{base}$, mT5$_{large}$, XGLM$_{564M}$, and XGLM$_{1.7B}$ on our 5-shot templated BLI data (XLING) and further study the effectiveness of our BLI-oriented fine-tuning (5K and 1K setups). The training set includes all XLING language pairs, where the 5K and 1K setups have $271, 754$ and $55, 228$ training instances respectively.

**Hyperparameters.** We first introduce our hyperparameters for BLI inference. In our main experiments, we adopt $n = 5$ [14] while in side experiments we further investigate and compare using different numbers of in-context examples $n$. Concerning the generation of output sequences, we adopt a beam size of 5 for all LLMs, and the maximum sequence length is 5 for encoder-decoder models and 5 plus the input sequence length for decoder-only models which first repeat the input sequence before generating new content. As for encoder-decoder LLMs, we use an evaluation batch size of 100 for smaller models and 8 for larger models as listed in Table 8 (Appendix B). Since the pretraining of decoder-only LLMs usually does not see padding tokens, we adopt a batch size of 1.[15] Following prior work (Li et al., 2022a,b), all our hyperparameters are tuned on (German, French), a randomly selected language pair.

For 'BLI-oriented' fine-tuning, we use the XLING data combining all language pairs, and the batch size is 16 for XGLM$_{1.7B}$ and 32 for mT5$_{base,large}$ and XGLM$_{564M}$. We use AdamW (Loshchilov and Hutter, 2019) with betas $= (0.9, 0.98)$ and a weight decay of 0.1. The learning rate is $2e$-6 for mT5$_{base}$, $1e$-6 for mT5$_{large}$, and $5e$-8 for XGLM$_{564M}$; concerning XGLM$_{1.7B}$, $5e$-9 is adopted for the 5K setup and $2e$-8 for the 1K setup. All the hyperparameters are tuned on

---

[10]However, in the appendix, we also provide the results with the remaining 3 languages: Croatian (HR), Finnish (FI), and Turkish (TR).

[11]Related work often also refers to the 5K and 1K cases as supervised and semi-supervised BLI setups, respectively (Li et al., 2022a).

[12]A summary concerning the detailed information of each mLLM is available in Appendix B.

[13]Experiments on BLI-oriented fine-tuning for a selection of mLLMs are discussed later.

[14]Unless otherwise stated, we report 5-shot BLI results throughout our experiments.

[15]We found that a larger batch size may cause a drop in the BLI performance.

the same randomly chosen language pair (German, French). Each LLM is fine-tuned for at most 20 epochs capped at 12 hours on $1\times$80GB A100 GPU.

**Baselines.** We adopt the following two SotA CLWE-based approaches as our baselines; both are open-source. We follow their original suggested hyperparameter choices respectively for 5K (supervised), 1K (semi-supervised), and unsupervised BLI setups, and we re-verify that the hyperparameters recommended are (near-)optimal. The Cross-domain Similarity Local Scaling (CSLS) retrieval (Lample et al., 2018) is adopted as recommended in the baselines.

• VECMAP (Artetxe et al., 2018a) is one of the most representative BLI approaches based on static CLWEs. It induces fastText-based CLWEs in various BLI supervision setups, and is notable for its effective self-learning mechanism, especially in weakly supervised and unsupervised BLI setups.

• CONTRASTIVEBLI (Li et al., 2022a) refines CLWEs with a two-stage contrastive learning procedure and reports the currently highest CLWE-based BLI scores on XLING and PanLex-BLI in 5K and 1K BLI setups. We adopt its strongest CLWEs derived with both fastText and mBERT (Devlin et al., 2019). CONTRASTIVEBLI does not support unsupervised BLI.

**BLI Evaluation.** Following previous work, we report the standard Precision@1 (P@1) scores both for our methods and for baseline methods.[16]

## 5 Results and Discussion

### 5.1 Main Results

**Comparison between mLLMs.** We compare the average BLI scores on 20 XLING BLI directions derived from all our 18 models from 5 LLM families in Figure 1. In all (5K, 1K, and zero-shot) BLI setups, the same general trends are observed. **1)** As expected, within the same mLLM family, larger models usually present stronger BLI capabilities, although exceptional cases exist (e.g., $XGLM_{7.5B}$ underperforms $XGLM_{4.5B}$). **2)** For encoder-decoder models, we find that mT5 outperforms mT0, showing that the instruction fine-tuning of mT0 does not benefit BLI in our experimental setups; **3)** LLaMA models achieve the strongest BLI performance among our 5 model families.

---

[16]P@1 is the most authoritative metric for BLI. Other measures such as P@5 and Mean Reciprocal Rank (MRR) show similar trends (Lample et al., 2018; Li et al., 2022a).

| [5K Setup] | VECMAP | CONTRASTIVEBLI | mT5$_{xxl}$ | mT0$_{xxl}$ | XGLM$_{4.5B}$ | mGPT | LLaMA$_{13B}$ |
|---|---|---|---|---|---|---|---|
| DE→ * | 47.65 | 54.02 | 49.1 | 35.73 | 48.43 | 37.44 | **56.44** |
| * →DE | 47.34 | 53.64 | 46.37 | 32.66 | 46.56 | 32.47 | **54.78** |
| EN→ * | 53.54 | 60.26 | 59.36 | 44.2 | 61.2 | 45.69 | **69.0** |
| * →EN | 57.38 | 60.97 | 57.34 | 41.48 | 56.08 | 45.15 | **62.35** |
| FR→ * | 53.36 | 58.51 | 53.46 | 39.7 | 54.31 | 44.02 | **60.81** |
| * →FR | 56.74 | 60.83 | 57.29 | 41.9 | 57.12 | 43.97 | **65.18** |
| IT→ * | 53.52 | 58.6 | 52.34 | 36.73 | 51.37 | 40.57 | **58.66** |
| * →IT | 55.61 | 59.88 | 55.43 | 40.67 | 55.86 | 47.06 | **64.4** |
| RU→ * | 46.74 | 52.13 | 49.88 | 34.09 | 50.08 | 38.25 | **56.26** |
| * →RU | 37.74 | 48.21 | 47.72 | 33.75 | 49.76 | 37.33 | **54.45** |
| Avg. | 50.96 | 56.71 | 52.83 | 38.09 | 53.08 | 41.2 | **60.23** |
| **[1K Setup]** | **VECMAP** | **CONTRASTIVEBLI** | **mT5$_{xxl}$** | **mT0$_{xxl}$** | **XGLM$_{4.5B}$** | **mGPT** | **LLaMA$_{13B}$** |
| DE→ * | 44.44 | 51.79 | 46.02 | 33.57 | 46.9 | 35.98 | **53.82** |
| * →DE | 44.0 | 48.92 | 43.62 | 30.58 | 44.06 | 31.9 | **53.36** |
| EN→ * | 47.7 | 55.11 | 56.19 | 40.01 | 59.3 | 44.55 | **68.27** |
| * →EN | 55.74 | **59.66** | 52.86 | 39.6 | 53.51 | 43.18 | 57.06 |
| FR→ * | 48.47 | 55.33 | 49.46 | 37.03 | 51.08 | 42.82 | **58.35** |
| * →FR | 54.88 | 58.65 | 54.51 | 38.88 | 56.74 | 43.9 | **63.26** |
| IT→ * | 49.08 | 55.92 | 48.49 | 33.78 | 49.97 | 39.86 | **56.06** |
| * →IT | 53.4 | 57.08 | 51.75 | 38.15 | 54.19 | 45.38 | **62.57** |
| RU→ * | 43.61 | 50.17 | 46.69 | 32.71 | 48.67 | 37.46 | **52.75** |
| * →RU | 25.3 | 44.02 | 44.1 | 29.9 | 47.42 | 36.31 | **53.01** |
| Avg. | 46.66 | 53.66 | 49.37 | 35.42 | 51.18 | 40.13 | **57.85** |
| **[Zero-Shot]** | **VECMAP** | **CONTRASTIVEBLI** | **mT5$_{xxl}$** | **mT0$_{xxl}$** | **XGLM$_{4.5B}$** | **mGPT** | **LLaMA$_{13B}$** |
| DE→ * | **44.44** | - | 35.76 | 29.31 | 38.37 | 22.65 | 43.3 |
| * →DE | 43.95 | - | 42.4 | 41.75 | 40.76 | 21.36 | **47.53** |
| EN→ * | 47.76 | - | 50.94 | 36.48 | 51.01 | 28.6 | **54.44** |
| * →EN | **55.8** | - | 43.16 | 38.77 | 44.75 | 25.78 | 52.34 |
| FR→ * | 48.24 | - | 42.37 | 35.66 | 45.24 | 30.73 | **51.01** |
| * →FR | **54.96** | - | 46.04 | 32.16 | 45.79 | 32.49 | 53.87 |
| IT→ * | **48.97** | - | 38.84 | 30.97 | 39.08 | 27.4 | 48.04 |
| * →IT | **53.27** | - | 45.68 | 37.4 | 46.81 | 34.79 | 53.09 |
| RU→ * | 43.63 | - | 42.05 | 32.19 | 40.85 | 21.49 | **46.82** |
| * →RU | 25.08 | - | 32.69 | 24.5 | 36.45 | 16.46 | **36.76** |
| Avg. | 46.61 | - | 41.99 | 32.92 | 42.91 | 26.18 | **48.72** |

Table 1: Main results on 20 XLING BLI directions in 5K, 1K, and zero-shot (unsupervised) setups. Off-the-shelf mLLMs are used without any fine-tuning. Average P@1$\times$100% scores of each language going to and from other 4 languages are reported. '-': CONTRASTIVEBLI does not support unsupervised BLI.

**XLING: Main Results.** In Table 1, we report the BLI performance of the strongest single model from each LLM family (full results for each of the 18 LLMs are available in Appendix D). Our results on 20 XLING BLI directions reaffirm the leading position of LLaMA where the LLaMA$_{13B}$ variant achieves the highest overall average BLI scores in all BLI setups, also outperforming CONTRASTIVEBLI, the previous CLWE-based SotA on the same dataset. We speculate that LLMs are more adept at few-shot learning: LLaMA$_{13B}$ outperforms VECMAP by circa 10 P@1 points in few-shot setups, but only by about 2 points in the zero-shot setup. It is also worth mentioning that in 5K and 1K setups, mT5$_{xxl}$ and XGLM$_{4.5B}$ beat VECMAP although they underperform CONTRASTIVEBLI; however, in the zero-shot setup they still cannot match VECMAP.

**PanLex-BLI: Main Results.** We present our results on lower-resource languages from PanLex-BLI in Table 2. We also provide here only one strongest model of each LLM family, while the full results from all 18 LLMs are available in Appendix D). This time, LLaMA$_{13B}$ still outperforms other LLMs, but we report here XGLM$_{7.5B}$ instead which beats XGLM$_{4.5B}$. Unlike for XLING, we

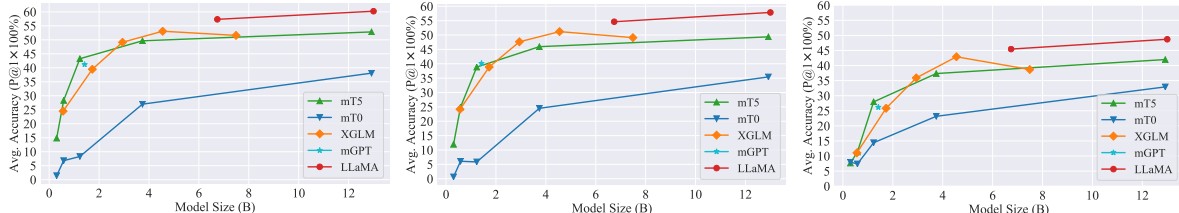

Figure 1: Averaged BLI score versus model size (0.3B to 13B): (left) $|\mathcal{D}_S|$=5K; (middle) $|\mathcal{D}_S|$=1K; (right) $|\mathcal{D}_S|$=0.

find that traditional CLWE-based approaches still outperform LLM-elicited BLI in general. This may reveal that current SotA mLLMs (size≤13B) still lack strong word translation capabilities for a large number of languages and language pairs, even for those they currently cover.

Put simply, while current mLLMs do exhibit strong performance for arguably high-resource languages (from XLING), they still have deficiencies with lower-resource languages as well as with their portability to a much larger number of languages, currently covered by more traditional BLI approaches (Li et al., 2022a). We leave to future work the investigation of larger mLLMs (e.g., LLaMA$_{30B}$) for BLI with lower-resource languages and languages unseen by the mLLMs.

**Statistical Significance.** We conduct $\chi^2$ test comparing LLaMA$_{13B}$ against the strongest single baseline in each BLI setup (i.e., CONTRASTIVEBLI in few-shot setups and VECMAP in the zero-shot setup) on the average BLI performance over 20 XLING and 6 PanLex-BLI BLI directions respectively, and we estimate the $p$-values as follows. **1)** On XLING, $p$ is $2.8e\text{-}23$ in the 5K setup, $8.5e\text{-}32$ in the 1K setup, and $4.3e\text{-}9$ in the zero-shot setup. **2)** For PanLex-BLI, $p$ is $1e\text{-}4$ in the 1K setup and $1.9e\text{-}35$ in the zero-shot setup. The $p$-values show that our main findings are clearly statistically significant.[17]

## 5.2 Further Analyses

$n$-**Shot Prompting.** To better understand the influence of the number of in-context examples, we pick mT5$_{large}$ (an encoder-decoder LLM) and LLaMA$_{13B}$ (a decoder-only LLM) and run experiments ranging from 0-shot to 10-shot. Figure 2 depicts their average BLI scores on 20 XLING BLI directions in 5K and 1K setups, respectively. The results clearly demonstrate the usefulness of in-context learning. Even when having only one in-context example (one-shot), the same model vari-

[17] By convention, $p < 0.05$: statistically significant; $p < 1e\text{-}3$: statistically highly significant.

| [1K Setup] | VECMAP | CONTRASTIVEBLI | mT5$_{xxl}$ | mT0$_{xxl}$ | XGLM$_{7.5B}$ | mGPT | LLaMA$_{13B}$ |
|---|---|---|---|---|---|---|---|
| BG→CA | 39.66 | **43.93** | 38.67 | 31.72 | 40.19 | - | 41.71 |
| CA→BG | 33.54 | 40.06 | 36.2 | 22.72 | 40.23 | - | **41.53** |
| BG→HU | 38.77 | **44.62** | 36.17 | 25.46 | - | 23.61 | 36.57 |
| HU→BG | 36.52 | 43.03 | 36.98 | 24.02 | - | 28.17 | **43.2** |
| CA→HU | 35.47 | **41.44** | 32.43 | 22.21 | - | - | 35.3 |
| HU→CA | 39.88 | **47.14** | 37.68 | 29.59 | - | - | 46.04 |
| Avg. | 37.31 | **43.37** | 36.36 | 25.95 | - | - | 40.72 |

| [Zero-Shot] | VECMAP | CONTRASTIVEBLI | mT5$_{xxl}$ | mT0$_{xxl}$ | XGLM$_{7.5B}$ | mGPT | LLaMA$_{13B}$ |
|---|---|---|---|---|---|---|---|
| BG→CA | **39.6** | - | 28.04 | 28.86 | 28.5 | - | 32.77 |
| CA→BG | **33.6** | - | 21.47 | 16.83 | 20.17 | - | 27.03 |
| BG→HU | **39.24** | - | 27.26 | 24.07 | - | 7.23 | 23.61 |
| HU→BG | **36.46** | - | 22.47 | 16.94 | - | 9.85 | 26.5 |
| CA→HU | **34.09** | - | 24.59 | 22.93 | - | - | 24.53 |
| HU→CA | 37.79 | - | 25.47 | 24.48 | - | - | **38.17** |
| Avg. | **36.8** | - | 24.88 | 22.35 | - | - | 28.77 |

Table 2: Main results on 6 PanLex-BLI BLI directions in 1K and zero-shot (unsupervised) setups. Off-the-shelf mLLMs are used without any fine-tuning. P@1×100% scores are reported. '-': **1)** a language is not supported by the LLM; **2)** CONTRASTIVEBLI does not support unsupervised BLI.

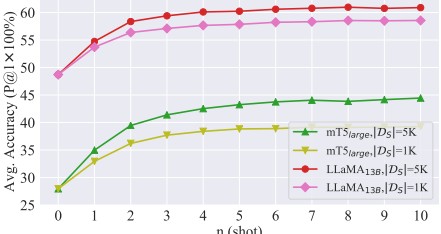

Figure 2: BLI scores averaged over 20 BLI directions from XLING with respect to the number of in-context examples $n$ (0 to 10), with mT5$_{large}$ and LLaMA$_{13B}$ in both 5K and 1K BLI setups.

ant outperforms its zero-shot results by $\sim 5$ P@1 points. However, with higher values for $n$ (i.e., $n \geq 5$), the gains become saturated.

**Ablation Study.** One key contribution of our work is that we propose to extract in-context examples from nearest neighbours. To validate the effectiveness of this approach, we conduct an essential ablation study where we use randomly sampled in-context examples instead. As with our main experiments, we present average scores on 20 BLI directions from only one best LLM from each LLM family, and full results on all LLMs are available in Appendix D. Our results in Table 3 demonstrate the following. **1)** The nearest neighbour-based 'NN (*K)' scores for every LLM outperform the 'Ran-

dom (*K)' scores by a salient margin: this shows the effectiveness of in-context examples from nearest neighbours. **2)** 'Random (*K)' outperforms 'Zero-Shot', showing that even randomly picked in-context examples can benefit BLI. **3)** 'NN (5K)' outperforms 'NN (1K)', which means that better in-context examples can be retrieved from a larger (and more varied) 5K seed dictionary.[18] We further show that these findings are statistically significant via $\chi^2$ test and report the $p$-values in Table 4.

|  | mT5$_{xxl}$ | mT0$_{xxl}$ | XGLM$_{4.5B}$ | mGPT | LLaMA$_{13B}$ |
|---|---|---|---|---|---|
| NN (5K) | 52.83 | 38.09 | 53.08 | 41.2 | 60.23 |
| NN (1K) | 49.37 | 35.42 | 51.18 | 40.13 | 57.85 |
| Random (5K) | 46.93 | 34.97 | 49.85 | 38.88 | 56.85 |
| Random (1K) | 45.93 | 33.90 | 49.92 | 38.38 | 56.11 |
| Zero-Shot (Unsupervised) | 41.99 | 32.92 | 42.91 | 26.18 | 48.72 |

Table 3: Ablation results. Averaged BLI scores (P@1×100%) on 20 XLING BLI directions. Rows 1-2: 5-shot prompting with in-context examples extracted from NN in $\mathcal{D}_S$ of size 5K and 1K. Rows 3-4: 5-shot prompting with random in-context examples in $\mathcal{D}_S$ of size 5K and 1K. Row 5: zero-shot prompting without any in-context examples.

| $p$-value | mT5$_{xxl}$ | mT0$_{xxl}$ | XGLM$_{4.5B}$ | mGPT | LLaMA$_{13B}$ |
|---|---|---|---|---|---|
| NN (5K) vs. Random (5K) | 1.3e-60 | 1.9e-19 | 2.4e-19 | 4.7e-11 | 1.41e-21 |
| NN (1K) vs. Random (1K) | 9.3e-22 | 9.1e-6 | 4.6e-4 | 6.4e-7 | 1.1e-6 |
| Random (5K) vs. Zero-Shot | 1.7e-43 | 1.8e-9 | 1.4e-83 | 1e-311 | 8.4e-114 |
| Random (1K) vs. Zero-Shot | 2.4e-28 | 3.9e-3 | 3.2e-85 | 5.8e-289 | 2.8e-94 |
| NN (5K) vs. NN (1K) | 5.8e-22 | 1.3e-14 | 1.2e-7 | 2.5e-3 | 1.7e-11 |

Table 4: Statistical significance associated with Table 3. We conduct $\chi^2$ tests and report $p$-values.

**BLI-Oriented Fine-Tuning.** The fine-tuning experiments, due to computational constraints, are conducted on four relatively smaller-scale LLMs. Table 5 reports each model's average performance on 20 XLING BLI directions before and after fine-tuning. We run fine-tuning experiments three times with different random seeds and report both mean scores and standard deviations. We only observe salient gains on mT5 models and XGLM$_{564M}$ in the 5K setups. For XGLM$_{1.7B}$ in both setups and all models in the 1K setup, the gains are smaller or even non-existent. Even in the 5K setup, the tuned mT5$_{base}$ still cannot match the off-the-shelf mT5$_{large}$, and the tuned mT5$_{large}$ underperforms mT5$_{xl}$ (cf. Table 18 in Appendix D). This may indicate some of the limitations of our proposed

---

|  | mT5$_{base}$ | mT5$_{large}$ | XGLM$_{564M}$ | XGLM$_{1.7B}$ |
|---|---|---|---|---|
| Fine-Tuned (5K) | **36.11**±0.531 | **46.68**±0.058 | **28.94**±0.026 | **40.92**±0.005 |
| Off-The-Shelf (5K) | 28.33 | 43.25 | 24.51 | 39.49 |
| Fine-Tuned (1K) | **25.61**±0.307 | 38.48±0.193 | **25.6**±0.123 | **39.59**±0.017 |
| Off-The-Shelf (1K) | 24.91 | **38.84** | 24.18 | 38.86 |

Table 5: Comparisons between mLLMs before and after BLI-oriented fine-tuning. Averaged BLI scores (P@1×100%) on 20 XLING BLI directions.

BLI-oriented fine-tuning (i.e., training the mLLMs on BLI data with their own pretraining objectives) and may indicate the following. **1)** With the current training approach and a fixed amount of computational budget, one may prioritise adopting off-the-shelf larger LLMs (with in-context learning) rather than training smaller-scale ones. **2)** In future work, other training objectives and strategies should be investigated for improved BLI performance with mLLMs. As another future research avenue, it is also worth extending the training to larger mLLMs and adopting novel fine-tuning techniques such as prompt tuning (Lester et al., 2021), adapters (Li et al., 2020, 2023) and LoRA (Hu et al., 2022).

**Templates.** Now, we additionally provide some preliminary findings from our template search.[19] **1)** Models from the same mLLM family may tend to prefer the same template. For example, Table 12 (Appendix C) shows that all five XGLM models prefer the same best zero-shot template and four of them share one best few-shot template. This phenomenon is to some extent seen also on mT5 (zero-shot and few-shot), mT0 (zero-shot and few-shot), and LLaMA (few-shot). This should be due to the same training data, training strategy, and model architecture adopted for all models in the same LLM family. **2)** As already mentioned in §3.1, mT0 is compatible with both mask-filling-style and GPT-style templates: Table 12 shows that some mT0 models prefer templates with '<mask>' and others do not. **3)** Under the 'GPT-style' templates, decoder-only models all prefer templates for sentence completion while some of the instruction-tuned mT0 models prefer questions with '?'.

### 5.3 Further Discussion

**Few-Shot Learning for BLI.** Our main results demonstrate that few-shot learning derives consistent gains over zero-shot prompting. For instance, HR→EN and IT→EN saw 345 and 272 cases in

---

[18]We also observe a slight edge of 'Random (5K)' over 'Random (1K)', and we speculate this might have to do with how XLING's test set and seed dictionaries were created. In fact, the 1K seed dictionary contains the most frequent 1K words, and the test set words include less frequent 2K words.

[19]Since we conduct template search only on a random language pair, these findings are yet to be verified by future work for other language pairs.

their test sets respectively where few-shot learning makes the correct prediction but zero-shot learning fails (positive cases). There are only 85 and 87 cases where zero-shot prompting beats few-shot prompting (negative cases). We present 8 positive examples and 4 negative examples for each of HR→EN and IT→EN, comparing five-shot (5K setup) and zero-shot results with LLaMA$_{13B}$ in Table 19 (Appendix E). For instance, 'gušter (HR) → lizard (EN)' and 'sezam (HR) → sesame' are two positive cases, their in-context examples being five different animal names and five plant names, which may help LLaMA$_{13B}$ to narrow down the scope of the target word to animal and plant names respectively. Similarly, 'valcer (HR) → waltz (EN)' (a positive case) is associated with five in-context examples related to either music or dance. However, few-shot learning does not always help. For example, in 'eventuale (IT) → eventual (EN)' and 'scopre (IT) → discovers (EN)' translation tasks, the LLM seems to make a mistake due to directly copying one of the words provided in the in-context examples, whereas zero-shot prompting predicts the correct answers.

**BLI for EN and non-EN Languages.** It is noteworthy that the volume of data in English for mLLM pretraining often exceeds that in any other language (e.g., mT5, mT0, and LLaMA), and thus mLLMs may be biased, favouring BLI directions involving EN. However, we did not identify very clear clues indicating that their BLI performance is (heavily) biased. In fact, in the 5K setup (see Table 1), although 'EN→∗' surpasses 'non-EN→∗' in absolute BLI scores (for each of our LLMs and also CLWE-based baselines), we meanwhile observe that **1)** mT0$_{xxl}$ achieves lower average score in '∗→EN' than '∗→FR', and **2)** for LLaMA$_{13B}$, '∗→EN' lags behind both '∗→FR' and '∗→IT'. Moreover, as an example, the LLaMA$_{13B}$ model supports 6 languages resulting in 30 BLI directions, and 20 of them are between non-EN languages. LLaMA$_{13B}$ outperforms CLWE-based SotA in 16/20 cases and in 18/20 cases respectively in the 5K and 1K setups for the non-EN pairs (cf. Tables 13 and 14). However, we again note that this might hold only for high-resource languages such as the ones covered in XLING.

**Impact Statement.** Here, we discuss the potential impact of our study on the following two aspects. **1)** *On future BLI research.* Our work minimises the technical gap between BLI and prompt-based learning and opens up new possibilities for BLI research. In fact, LLM prompting provides a generic and straightforward way of leveraging external knowledge for BLI. While we have demonstrated the effectiveness of in-context word translation examples, external information such as word definition, parts-of-speech, spelling, and sentence translation pairs can also be integrated into text prompts. **2)** *On NMT and other related fields.* Recent work has incorporated word translation pairs into text templates to prompt LLMs for sentence-level neural machine translation (NMT) and demonstrates that the bilingual lexical 'hints' lead to significant gains in NMT (Ghazvininejad et al., 2023; Jones et al., 2023). While a ground-truth bilingual dictionary can be leveraged, BLI is able to provide word translations for language pairs and words not covered in existing bilingual lexica.[20] Our work can provide strong word translation pairs for lexicon-enhanced MT, and the improved MT may further benefit, e.g., the field of cross-lingual transfer learning via TRANSLATE-TRAIN/TEST approaches (Conneau et al., 2018; Li et al., 2023).

# 6 Conclusion

This paper presents the first study on bilingual lexicon induction (BLI) with multilingual text-to-text large language models (mLLMs). We develop the methodology to prompt mLLMs for BLI, conduct extensive template search, and systematically experiment with 5 representative mLLM families (18 models) on a variety of zero-shot and few-shot BLI tasks. Relying on off-the-shelf mLLMs, our experiments on the standard XLING dataset offer strong performance in all BLI setups, where our proposed few-shot prompting with in-context examples from nearest neighbours outperforms the strongest CLWE-based SotA by a considerable margin. However, our study also points out that prompting-based methods still need to be successfully extended to lower-resource languages. Finally, we conduct a series of in-depth analyses covering variants of our few-shot prompting and preliminary investigations on BLI-oriented fine-tuning. Our key findings and comprehensive analyses may pave the way for the development of stronger mLLM-based BLI systems in the future.

---

[20]Moreover, given an input word, BLI can offer multiple plausible translations for the downstream NMT to consider, and we speculate this may, to some extent, increase the diversity of MT output.

## Limitations

First, most recently released state-of-the-art mLLMs are still unable to support as many languages as static word embeddings, which currently limits their wider portability. For instance, LLaMA supports 20 languages and XGLM supports 30 languages, while fastText provides pretrained static WEs for 294 languages that can be used for the induction of static CLWEs.[21] Intuitively, this is because training LLMs that support more languages would require higher computational costs (with more training data and typically larger model sizes). We hope that researchers in the future can pretrain and release mLLMs that support a larger set of linguistically diverse languages, which can thus probably extend the success of our approach to more languages and language families.

Second, our work did not investigate open-source LLMs with more than 13B parameters[22] due to a large number of experiments conducted combined with our limited computing resources, and we did not evaluate any closed-source LLMs. Quite a few tech companies and AI research labs have been training LLMs with 100+B and even 500+B parameters. We encourage interested readers who have access to adequate computing resources or specific closed-source LLMs to take a step further and investigate if larger LLMs can provide an even stronger BLI performance than reported in this particular work, following the recipe presented in this work.

Third, as also discussed in other BLI work (Li et al., 2022b), existing BLI datasets did not control the synonyms and polysemy well and to a sufficient detail. In fact, when constructing BLI datasets, it is very difficult to collect all correct translations for each source word. Therefore, one limitation of BLI evaluation is that it cannot give credit to correct answers that are not included in the ground-truth translation set, and evaluation is typically conducted out-of-context. Constructing finer-grained BLI datasets with the help of qualified annotators (e.g., linguists, typologists and bilingual speakers) is beyond the scope of this work.

## Acknowledgements

We thank the anonymous reviewers and area chairs for their valuable feedback. Yaoyiran Li is supported by Grace & Thomas C. H. Chan Cambridge International Scholarship. Ivan Vulić is supported by a personal Royal Society University Research Fellowship *'Inclusive and Sustainable Language Technology for a Truly Multilingual World'* (no 221137; 2022–).

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

## A  Languages

| Family | Language | Code | LLMs |
|---|---|---|---|
| IE:Germanic | English | EN | mT5,mT0,XGLM,mGPT,LLaMA |
| | German | DE | mT5,mT0,XGLM,mGPT,LLaMA |
| IE:Romance | Catalan | CA | mT5,mT0,XGLM,LLaMA |
| | French | FR | mT5,mT0,XGLM,mGPT,LLaMA |
| | Italian | IT | mT5,mT0,XGLM,mGPT,LLaMA |
| IE:Slavic | Bulgarian | BG | mT5,mT0,XGLM,mGPT,LLaMA |
| | Croatian | HR | LLaMA |
| | Russian | RU | mT5,mT0,XGLM,mGPT,LLaMA |
| Turkic | Turkish | TR | mT5,mT0,XGLM,mGPT |
| Uralic | Finnish | FI | mT5,mT0,XGLM,mGPT |
| | Hungarian | HU | mT5,mT0,mGPT,LLaMA |

Table 6: Languages covered in our experiments with their ISO 639-1 codes and the mLLM families that support that language, categorized by language family. IE = Indo-European.

## B  Reproducibility Checklist

• **BLI Data**: We adopt two publicly available BLI datasets.[23] [24]

• **Static Word Embeddings**: Following the datasets' own recommendations and other previous work, we use the XLING-preprocessed fastText WEs trained on Wikipedia[25] for XLING data and fastText WEs trained on Common Crawl + Wikipedia[26] for PanLex-BLI, and the WEs are trimmed to the most frequent 200K words for each language. For fair comparisons, we use the same set of fastText WEs both for the retrieval of nearest neighbours (in-context examples) and for the CLWE-based baselines.

• **Pretrained LLMs and Parameter Counts**: All the LLMs used in our experiments are publicly available from the `huggingface.co` model hub. We summarise their model identifiers and model sizes in Table 7. Please refer to each LLM's own copyright and licence before downloading, using, fine-tuning, or redistributing any LLM.

• **Source Code**: Our code is publicly available at https://github.com/cambridgeltl/prompt4b li.

• **Computing Infrastructure**: We have run our code on Wilkes3, a GPU cluster hosted by Re-search Computing Services at the University of Cambridge, where each run leverages a single Nvidia 80GB A100 GPU and $32\times$ CPU cores.

• **Software**: Slurm 20.11.9, Python 3.9.7, PyTorch 1.10.1+cu113, Transformers 4.28.1.

• **Runtime (Wall Time)**: We present the average inference time on one single BLI direction (i.e., circa $2,000$ word pairs in an XLING test set; the time required for loading the LLM and the dataset is not included) for each LLM in Table 8. The per-epoch training time for BLI-oriented fine-tuning is provided in Table 9.

• **Hyperparameter Search**: As introduced in §3.3 and §4, our template selection and all our hyperparameter search are conducted on a single randomly chosen language pair (German, French), following previous work (Li et al., 2022a,b). The learning rate for LLM fine-tuning is selected from $[1e-9, 5e-9, 1e-8, 2e-8, 1e-7, 1e-6, 2e-6, 2e-5, 1e-3]$.

• **Significance**: We have discussed the significance of our main results and ablation results in the last paragraph of §5.1 and in Table 4 respectively, which demonstrates that our findings are statistically significant.

• **Randomness**: Our main experiments are completely deterministic since we rely on off-the-shelf LLMs without any fine-tuning, nearest neighbour retrieval for in-context examples (a deterministic retrieval algorithm), and the deterministic beam search. The randomness only exists in two parts of our side analysis. First, we use random in-context examples in our ablation study, and we verify our findings with statistical tests in Table 4. Second, the fine-tuning experiments do have randomness, and we run fine-tuning three times for each model, reporting both average BLI performance and the standard deviation.

• **Carbon Footprint**: All the experiments involved in this project including hyperparameter tuning, template search, BLI inference, and BLI-oriented fine-tuning of our LLMs consume circa $1,650$ A100 GPU hours. Based on a publicly available 'machine learning emissions calculator' (Luccioni et al., 2019)[27] and our computational infrastructure, we estimate that our work causes the emission of circa 200kg $CO_2$ equivalents.

---

[23] https://github.com/codogogo/xling-eval
[24] https://github.com/cambridgeltl/panlex-bli
[25] https://fasttext.cc/docs/en/pretrained-vectors.html
[26] https://fasttext.cc/docs/en/crawl-vectors.html

[27] https://mlco2.github.io/impact/#compute

| LLM | Model ID | Number of Parameters |
|---|---|---|
| $\text{mT5}_{\text{small}}$ | "google/mt5-small" | $300,176,768$ |
| $\text{mT5}_{\text{base}}$ | "google/mt5-base" | $582,401,280$ |
| $\text{mT5}_{\text{large}}$ | "google/mt5-large" | $1,229,581,312$ |
| $\text{mT5}_{\text{xl}}$ | "google/mt5-xl" | $3,742,619,648$ |
| $\text{mT5}_{\text{xxl}}$ | "google/mt5-xxl" | $12,921,057,280$ |
| $\text{mT0}_{\text{small}}$ | "bigscience/mt0-small" | $300,176,768$ |
| $\text{mT0}_{\text{base}}$ | "bigscience/mt0-base" | $582,401,280$ |
| $\text{mT0}_{\text{large}}$ | "bigscience/mt0-large" | $1,229,581,312$ |
| $\text{mT0}_{\text{xl}}$ | "bigscience/mt0-xl" | $3,742,619,648$ |
| $\text{mT0}_{\text{xxl}}$ | "bigscience/mt0-xxl" | $12,921,057,280$ |
| $\text{XGLM}_{\text{564M}}$ | "facebook/xglm-564M" | $564,463,616$ |
| $\text{XGLM}_{\text{1.7B}}$ | "facebook/xglm-1.7B" | $1,732,907,008$ |
| $\text{XGLM}_{\text{2.9B}}$ | "facebook/xglm-2.9B" | $2,941,505,536$ |
| $\text{XGLM}_{\text{4.5B}}$ | "facebook/xglm-4.5B" | $4,552,511,488$ |
| $\text{XGLM}_{\text{7.5B}}$ | "facebook/xglm-7.5B" | $7,492,771,840$ |
| mGPT | "sberbank-ai/mGPT" | $1,417,596,928$ |
| $\text{LLaMA}_{\text{7B}}$ | "huggyllama/llama-7b" | $6,738,415,616$ |
| $\text{LLaMA}_{\text{13B}}$ | "huggyllama/llama-13b" | $13,015,864,320$ |

Table 7: LLMs used in our experiments with their `huggingface.co` model IDs and model sizes.

| LLM | Batch Size (Inference) | 0-Shot | 5-Shot |
|---|---|---|---|
| $\text{mT5}_{\text{small}}$ | 100 | 6s | 7s |
| $\text{mT5}_{\text{base}}$ | 100 | 7s | 8s |
| $\text{mT5}_{\text{large}}$ | 100 | 8s | 12s |
| $\text{mT5}_{\text{xl}}$ | 8 | 45s | 50s |
| $\text{mT5}_{\text{xxl}}$ | 8 | 52s | 83s |
| $\text{mT0}_{\text{small}}$ | 100 | 5s | 6s |
| $\text{mT0}_{\text{base}}$ | 100 | 6s | 7s |
| $\text{mT0}_{\text{large}}$ | 100 | 8s | 14s |
| $\text{mT0}_{\text{xl}}$ | 8 | 46s | 50s |
| $\text{mT0}_{\text{xxl}}$ | 8 | 57s | 78s |
| $\text{XGLM}_{\text{564M}}$ | 1 | 213s | 225s |
| $\text{XGLM}_{\text{1.7B}}$ | 1 | 228s | 237s |
| $\text{XGLM}_{\text{2.9B}}$ | 1 | 343s | 366s |
| $\text{XGLM}_{\text{4.5B}}$ | 1 | 336s | 394s |
| $\text{XGLM}_{\text{7.5B}}$ | 1 | 382s | 461s |
| mGPT | 1 | 192s | 210s |
| $\text{LLaMA}_{\text{7B}}$ | 1 | 328s | 463s |
| $\text{LLaMA}_{\text{13B}}$ | 1 | 434s | 636s |

Table 8: Inference time (in seconds) of each LLM with 0-Shot and 5-Shot prompts respectively.

| LLM | Batch Size (Training) | 5K Setup | 1K Setup |
|---|---|---|---|
| $\text{mT5}_{\text{base}}$ | 32 | 17 min | 4 min |
| $\text{mT5}_{\text{large}}$ | 32 | 38 min | 8 min |
| $\text{XGLM}_{\text{564M}}$ | 32 | 24 min | 5 min |
| $\text{XGLM}_{\text{1.7B}}$ | 16 | 80 min | 15 min |

Table 9: Per-epoch training time (in minutes) of each LLM with 5-Shot prompts in 5K and 1K setups respectively.

## C Templates

We summarise all our zero-shot templates in Table 10 and few-shot templates in Table 11: these 102 templates constitute our 'template pool'. Each of Tables 10 and 11 is split into two parts for mask-filling-style and GPT-style templates respectively as introduced in §3.1 and §3.2. In addition, we list the best zero-shot template and the best few-shot template for each of our 18 LLMs in Table 12. Again, as already mentioned in §3.3, our template selection is conducted on a randomly chosen language pair (German, French) where for few-show templates, the in-context examples are derived from a seed dictionary of size 5K. While we do not have enough computational resources to calculate and do not have enough space to present the performance of each template for each LLM on each XLING BLI direction ($102 \times 18 \times 56 = 102,816$ scores), in the last paragraph of §5.2 we have discussed some preliminary findings only from our template search.

## D Full BLI Results

Here we present our full results on both XLING and PanLex-BLI. Table 13, 14, and 16 are our results on all 56 XLING BLI directions in 5K, 1K, and zero-shot (unsupervised) BLI setups respectively. Table 15 and 17 are results for PanLex-BLI lower-resource languages (6 BLI directions) in 1K and zero-shot (unsupervised) BLI setups. Note that an (m)LLM usually cannot support every language, and we use '-' to denote this scenario. Throughout this paper, our expression 'a language is not supported by an LLM' means that the language is not used for pretraining the LLM even if the LLM's tokeniser may still be able to tokenise possibly many input sentences in the language. Table 18 shows the full ablation results for each of our 18 mLLMs.

## E Translation Examples

To illustrate how few-shot learning improves BLI, we present some of our BLI results with $\text{LLaMA}_{\text{13B}}$ in Table 19 comparing five-shot and zero-shot prompting on HR→EN and IT→EN BLI test sets.

| | **Mask-Filling-Style Templates (Zero-Shot Prompting)** | | |
|---|---|---|---|
| 1 | The word '$w^x$' in $L^y$ is: <mask>. | 2 | The word $w^x$ in $L^y$ is: <mask>. |
| 3 | The word '$w^x$' in $L^y$ is: <mask> | 4 | The word $w^x$ in $L^y$ is <mask> |
| 5 | The $L^x$ word $w^x$ in $L^y$ is: <mask>. | 6 | The $L^x$ word $w^x$ in $L^y$ is <mask>. |
| 7 | The $L^x$ word '$w^x$' in $L^y$ is: <mask>. | 8 | The $L^x$ word '$w^x$' in $L^y$ is <mask>. |
| 9 | The $L^x$ word $w^x$ in $L^y$ is: <mask> | 10 | The $L^x$ word $w^x$ in $L^y$ is <mask> |
| 11 | The $L^x$ word '$w^x$' in $L^y$ is: <mask> | 12 | The $L^x$ word '$w^x$' in $L^y$ is <mask> |
| 13 | '$w^x$' in $L^y$ is: <mask>. | 14 | $w^x$ in $L^y$ is: <mask>. |
| 15 | '$w^x$' in $L^y$ is: <mask> | 16 | $w^x$ in $L^y$ is: <mask> |
| 17 | What is the translation of the word '$w^x$' into $L^y$? <mask>. | 18 | What is the translation of the word $w^x$ into $L^y$? <mask>. |
| 19 | What is the translation of the $L^x$ word '$w^x$' into $L^y$? <mask>. | 20 | What is the translation of the $L^x$ word $w^x$ into $L^y$? <mask>. |
| 21 | The translation of the word '$w^x$' into $L^y$ is <mask>. | 22 | The translation of the word $w^x$ into $L^y$ is <mask>. |
| 23 | The translation of the $L^x$ word '$w^x$' into $L^y$ is <mask>. | 24 | How do you say '$w^x$' in $L^y$? <mask>. |
| 25 | How do you say $w^x$ in $L^y$? <mask>. | 26 | How do you say the $L^x$ word '$w^x$' in $L^y$? <mask>. |
| 27 | How do you say the $L^x$ word $w^x$ in $L^y$? <mask>. | 28 | Translate the word '$w^x$' into $L^y$: <mask>. |
| 29 | Translate the word $w^x$ into $L^y$: <mask>. | 30 | Translate the word $w^x$ into $L^y$: <mask> |
| 31 | Translate $w^x$ into $L^y$: <mask>. | 32 | Translate the $L^x$ word $w^x$ into $L^y$: <mask>. |
| 33 | Translate the $L^x$ word $w^x$ into $L^y$: <mask> | 34 | Translate from $L^x$ to $L^y$: $w^x$-> <mask>. |
| 35 | Translate from $L^x$ to $L^y$: $w^x$-> <mask> | 36 | Translate from $L^x$ to $L^y$: $w^x$=> <mask>. |
| 37 | Translate from $L^x$ to $L^y$: $w^x$=> <mask> | | |
| | **GPT-Style Templates (Zero-Shot Prompting)** | | |
| 38 | The word '$w^x$' in $L^y$ is: | 39 | The word $w^x$ in $L^y$ is: |
| 40 | The word $w^x$ in $L^y$ is | 41 | The $L^x$ word $w^x$ in $L^y$ is: |
| 42 | The $L^x$ word $w^x$ in $L^y$ is | 43 | The $L^x$ word '$w^x$' in $L^y$ is: |
| 44 | The $L^x$ word '$w^x$' in $L^y$ is | 45 | '$w^x$' in $L^y$ is: |
| 46 | $w^x$ in $L^y$ is: | 47 | Translate the word '$w^x$' into $L^y$: |
| 48 | Translate the word $w^x$ into $L^y$: | 49 | Translate from $L^x$ to $L^y$: $w^x$-> |
| 50 | Translate from $L^x$ to $L^y$: $w^x$=> | 51 | Translate $w^x$ into $L^y$: |
| 52 | Translate the $L^x$ word $w^x$ into $L^y$: | 53 | Translate the $L^x$ word '$w^x$' into $L^y$: |
| 54 | What is the translation of the word '$w^x$' into $L^y$? | 55 | What is the translation of the word $w^x$ into $L^y$? |
| 56 | The translation of the word '$w^x$' into $L^y$ is | 57 | The translation of the word $w^x$ into $L^y$ is |
| 58 | The translation of the $L^x$ word '$w^x$' into $L^y$ is | 59 | The translation of the $L^x$ word $w^x$ into $L^y$ is |
| 60 | How do you say '$w^x$' in $L^y$? | 61 | How do you say $w^x$ in $L^y$? |
| 62 | How do you say '$w^x$' in $L^y$: | 63 | How do you say $w^x$ in $L^y$: |
| 64 | How do you say the $L^x$ word '$w^x$' in $L^y$? | 65 | How do you say the $L^x$ word $w^x$ in $L^y$? |
| 66 | Q: What is the $L^y$ translation of $w^x$ A: | | |

Table 10: Our 66 templates for zero-shot prompting. These include 37 mask-filling-style templates (template IDs: 1 ∼ 37) and 29 GPT-style templates (template IDs: 38 ∼ 66). In our experiments, the '<mask>' token is '<extra_id_0>' for mT5 and mT0.

| | **Mask-Filling-Style Templates (Few-Shot Prompting)** |
|---|---|
| 67 | Translate from $L^x$ to $L^y$: $w_1^x$->$w_1^y$ $w_2^x$->$w_2^y$ $w^x$-> <mask>. |
| 68 | Translate from $L^x$ to $L^y$: $w_1^x$->$w_1^y$, $w_2^x$->$w_2^y$, $w^x$-> <mask>. |
| 69 | Translate from $L^x$ to $L^y$: $w_1^x$->$w_1^y$ $w_2^x$->$w_2^y$ $w^x$-> <mask> |
| 70 | Translate from $L^x$ to $L^y$: $w_1^x$->$w_1^y$, $w_2^x$->$w_2^y$, $w^x$-> <mask> |
| 71 | Translate from $L^x$ to $L^y$: $w_1^x$=>$w_1^y$ $w_2^x$=>$w_2^y$ $w^x$=> <mask>. |
| 72 | Translate from $L^x$ to $L^y$: $w_1^x$=>$w_1^y$, $w_2^x$=>$w_2^y$, $w^x$=> <mask>. |
| 73 | Translate from $L^x$ to $L^y$: $w_1^x$=>$w_1^y$ $w_2^x$=>$w_2^y$ $w^x$=> <mask> |
| 74 | Translate from $L^x$ to $L^y$: $w_1^x$=>$w_1^y$, $w_2^x$=>$w_2^y$, $w^x$=> <mask> |
| 75 | The word $w_1^x$ in $L^y$ is $w_1^y$. The word $w_2^x$ in $L^y$ is $w_2^y$. The word $w^x$ in $L^y$ is <mask>. |
| 76 | The $L^x$ word $w_1^x$ in $L^y$ is $w_1^y$. The $L^x$ word $w_2^x$ in $L^y$ is $w_2^y$. The $L^x$ word $w^x$ in $L^y$ is <mask>. |
| 77 | The $L^x$ word '$w_1^x$' in $L^y$ is '$w_1^y$'. The $L^x$ word '$w_2^x$' in $L^y$ is '$w_2^y$'. The $L^x$ word '$w^x$' in $L^y$ is '<mask>'. |
| 78 | The $L^x$ word $w_1^x$ in $L^y$ is $w_1^y$, The $L^x$ word $w_2^x$ in $L^y$ is $w_2^y$, The $L^x$ word $w^x$ in $L^y$ is <mask>. |
| | **GPT-Style Templates (Few-Shot Prompting)** |
| 79 | Translate from $L^x$ to $L^y$: $w_1^x$->$w_1^y$ $w_2^x$->$w_2^y$ $w^x$-> |
| 80 | Translate from $L^x$ to $L^y$: $w_1^x$->$w_1^y$, $w_2^x$->$w_2^y$, $w^x$-> |
| 81 | Translate from $L^x$ to $L^y$: $w_1^x$=>$w_1^y$ $w_2^x$=>$w_2^y$ $w^x$=> |
| 82 | Translate from $L^x$ to $L^y$: $w_1^x$=>$w_1^y$, $w_2^x$=>$w_2^y$, $w^x$=> |
| 83 | The word $w_1^x$ in $L^y$ is $w_1^y$. The word $w_2^x$ in $L^y$ is $w_2^y$. The word $w^x$ in $L^y$ is |
| 84 | The word $w_1^x$ in $L^y$ is $w_1^y$. The word $w_2^x$ in $L^y$ is $w_2^y$. The word $w^x$ in $L^y$ is: |
| 85 | The $L^x$ word $w_1^x$ in $L^y$ is $w_1^y$. The $L^x$ word $w_2^x$ in $L^y$ is $w_2^y$. The $L^x$ word $w^x$ in $L^y$ is |
| 86 | The $L^x$ word $w_1^x$ in $L^y$ is $w_1^y$. The $L^x$ word $w_2^x$ in $L^y$ is $w_2^y$. The $L^x$ word $w^x$ in $L^y$ is: |
| 87 | The word $w_1^x$ in $L^y$ is $w_1^y$, The word $w_2^x$ in $L^y$ is $w_2^y$, The word $w^x$ in $L^y$ is |
| 88 | The word $w_1^x$ in $L^y$ is $w_1^y$, The word $w_2^x$ in $L^y$ is $w_2^y$, The word $w^x$ in $L^y$ is: |
| 89 | The $L^x$ word $w_1^x$ in $L^y$ is $w_1^y$, The $L^x$ word $w_2^x$ in $L^y$ is $w_2^y$, The $L^x$ word $w^x$ in $L^y$ is |
| 90 | The $L^x$ word $w_1^x$ in $L^y$ is $w_1^y$, The $L^x$ word $w_2^x$ in $L^y$ is $w_2^y$, The $L^x$ word $w^x$ in $L^y$ is: |
| 91 | The word '$w_1^x$' in $L^y$ is $w_1^y$. The word '$w_2^x$' in $L^y$ is $w_2^y$. The word '$w^x$' in $L^y$ is |
| 92 | The word '$w_1^x$' in $L^y$ is $w_1^y$. The word '$w_2^x$' in $L^y$ is $w_2^y$. The word '$w^x$' in $L^y$ is: |
| 93 | The $L^x$ word '$w_1^x$' in $L^y$ is $w_1^y$. The $L^x$ word '$w_2^x$' in $L^y$ is $w_2^y$. The $L^x$ word '$w^x$' in $L^y$ is |
| 94 | The $L^x$ word '$w_1^x$' in $L^y$ is $w_1^y$. The $L^x$ word '$w_2^x$' in $L^y$ is $w_2^y$. The $L^x$ word '$w^x$' in $L^y$ is: |
| 95 | The word '$w_1^x$' in $L^y$ is $w_1^y$, The word '$w_2^x$' in $L^y$ is $w_2^y$, The word '$w^x$' in $L^y$ is |
| 96 | The word '$w_1^x$' in $L^y$ is $w_1^y$, The word '$w_2^x$' in $L^y$ is $w_2^y$, The word '$w^x$' in $L^y$ is: |
| 97 | The $L^x$ word '$w_1^x$' in $L^y$ is $w_1^y$, The $L^x$ word '$w_2^x$' in $L^y$ is $w_2^y$, The $L^x$ word '$w^x$' in $L^y$ is |
| 98 | The $L^x$ word '$w_1^x$' in $L^y$ is $w_1^y$, The $L^x$ word '$w_2^x$' in $L^y$ is $w_2^y$, The $L^x$ word '$w^x$' in $L^y$ is: |
| 99 | The word $w_1^x$ in $L^y$ is $w_1^y$. The word $w_2^x$ in $L^y$ is $w_2^y$. How do you say $w^x$ in $L^y$? |
| 100 | The $L^x$ word $w_1^x$ in $L^y$ is $w_1^y$. The $L^x$ word $w_2^x$ in $L^y$ is $w_2^y$. How do you say the $L^x$ word $w^x$ in $L^y$? |
| 101 | The word '$w_1^x$' in $L^y$ is $w_1^y$. The word '$w_2^x$' in $L^y$ is $w_2^y$. How do you say '$w^x$' in $L^y$? |
| 102 | The $L^x$ word '$w_1^x$' in $L^y$ is $w_1^y$. The $L^x$ word '$w_2^x$' in $L^y$ is $w_2^y$. How do you say the $L^x$ word '$w^x$' in $L^y$? |

Table 11: Our 36 templates for few-shot prompting. For simplicity, we present only two in-context examples in each template. These include 12 mask-filling-style templates (template IDs: $67 \sim 78$) and 24 GPT-style templates (template IDs: $79 \sim 102$). In our experiments, the '<mask>' token is '<extra_id_0>' for mT5 and mT0.

| LLM | Best Template (Zero-Shot) |
| --- | --- |
| mT5$_{small}$ | The word '$w^x$' in $L^y$ is: <mask>. |
| mT5$_{base}$ | Translate the word '$w^x$' into $L^y$: <mask>. |
| mT5$_{large}$ | The $L^x$ word '$w^x$' in $L^y$ is: <mask>. |
| mT5$_{xl}$ | The $L^x$ word '$w^x$' in $L^y$ is: <mask>. |
| mT5$_{xxl}$ | The $L^x$ word '$w^x$' in $L^y$ is: <mask> |
| mT0$_{small}$ | Translate from $L^x$ to $L^y$: $w^x$=> <mask>. |
| mT0$_{base}$ | Translate from $L^x$ to $L^y$: $w^x$=> <mask>. |
| mT0$_{large}$ | Q: What is the $L^y$ translation of $w^x$ A: |
| mT0$_{xl}$ | How do you say the $L^x$ word '$w^x$' in $L^y$? |
| mT0$_{xxl}$ | How do you say the $L^x$ word '$w^x$' in $L^y$? |
| XGLM$_{564M}$ | The $L^x$ word $w^x$ in $L^y$ is: |
| XGLM$_{1.7B}$ | The $L^x$ word $w^x$ in $L^y$ is: |
| XGLM$_{2.9B}$ | The $L^x$ word $w^x$ in $L^y$ is: |
| XGLM$_{4.5B}$ | The $L^x$ word $w^x$ in $L^y$ is: |
| XGLM$_{7.5B}$ | The $L^x$ word $w^x$ in $L^y$ is: |
| mGPT | Translate the $L^x$ word $w^x$ into $L^y$: |
| LLaMA$_{7B}$ | The $L^x$ word $w^x$ in $L^y$ is: |
| LLaMA$_{13B}$ | Translate from $L^x$ to $L^y$: $w^x$=> |

| LLM | Best Template (Few-Shot) |
| --- | --- |
| mT5$_{small}$ | The word $w_1^x$ in $L^y$ is $w_1^y$. The word $w_2^x$ in $L^y$ is $w_2^y$. The word $w^x$ in $L^y$ is <mask>. |
| mT5$_{base}$ | The word $w_1^x$ in $L^y$ is $w_1^y$. The word $w_2^x$ in $L^y$ is $w_2^y$. The word $w^x$ in $L^y$ is <mask>. |
| mT5$_{large}$ | The $L^x$ word $w_1^x$ in $L^y$ is $w_1^y$. The $L^x$ word $w_2^x$ in $L^y$ is $w_2^y$. The $L^x$ word $w^x$ in $L^y$ is <mask>. |
| mT5$_{xl}$ | The $L^x$ word $w_1^x$ in $L^y$ is $w_1^y$, The $L^x$ word $w_2^x$ in $L^y$ is $w_2^y$, The $L^x$ word $w^x$ in $L^y$ is <mask>. |
| mT5$_{xxl}$ | The $L^x$ word $w_1^x$ in $L^y$ is $w_1^y$, The $L^x$ word $w_2^x$ in $L^y$ is $w_2^y$, The $L^x$ word $w^x$ in $L^y$ is <mask>. |
| mT0$_{small}$ | The word '$w_1^x$' in $L^y$ is $w_1^y$. The word '$w_2^x$' in $L^y$ is $w_2^y$. How do you say '$w^x$' in $L^y$? |
| mT0$_{base}$ | The $L^x$ word $w_1^x$ in $L^y$ is $w_1^y$. The $L^x$ word $w_2^x$ in $L^y$ is $w_2^y$. The $L^x$ word $w^x$ in $L^y$ is |
| mT0$_{large}$ | The $L^x$ word '$w_1^x$' in $L^y$ is '$w_1^y$'. The $L^x$ word '$w_2^x$' in $L^y$ is '$w_2^y$'. The $L^x$ word '$w^x$' in $L^y$ is '<mask>'. |
| mT0$_{xl}$ | The $L^x$ word $w_1^x$ in $L^y$ is $w_1^y$. The $L^x$ word $w_2^x$ in $L^y$ is $w_2^y$. The $L^x$ word $w^x$ in $L^y$ is: |
| mT0$_{xxl}$ | The $L^x$ word $w_1^x$ in $L^y$ is $w_1^y$, The $L^x$ word $w_2^x$ in $L^y$ is $w_2^y$, The $L^x$ word $w^x$ in $L^y$ is: |
| XGLM$_{564M}$ | The word $w_1^x$ in $L^y$ is $w_1^y$. The word $w_2^x$ in $L^y$ is $w_2^y$. The word $w^x$ in $L^y$ is |
| XGLM$_{1.7B}$ | The $L^x$ word $w_1^x$ in $L^y$ is $w_1^y$. The $L^x$ word $w_2^x$ in $L^y$ is $w_2^y$. The $L^x$ word $w^x$ in $L^y$ is |
| XGLM$_{2.9B}$ | The $L^x$ word $w_1^x$ in $L^y$ is $w_1^y$. The $L^x$ word $w_2^x$ in $L^y$ is $w_2^y$. The $L^x$ word $w^x$ in $L^y$ is |
| XGLM$_{4.5B}$ | The $L^x$ word $w_1^x$ in $L^y$ is $w_1^y$. The $L^x$ word $w_2^x$ in $L^y$ is $w_2^y$. The $L^x$ word $w^x$ in $L^y$ is |
| XGLM$_{7.5B}$ | The $L^x$ word $w_1^x$ in $L^y$ is $w_1^y$. The $L^x$ word $w_2^x$ in $L^y$ is $w_2^y$. The $L^x$ word $w^x$ in $L^y$ is |
| mGPT | The $L^x$ word $w_1^x$ in $L^y$ is $w_1^y$. The $L^x$ word $w_2^x$ in $L^y$ is $w_2^y$. The $L^x$ word $w^x$ in $L^y$ is |
| LLaMA$_{7B}$ | The $L^x$ word '$w_1^x$' in $L^y$ is $w_1^y$. The $L^x$ word '$w_2^x$' in $L^y$ is $w_2^y$. The $L^x$ word '$w^x$' in $L^y$ is |
| LLaMA$_{13B}$ | The $L^x$ word '$w_1^x$' in $L^y$ is $w_1^y$. The $L^x$ word '$w_2^x$' in $L^y$ is $w_2^y$. The $L^x$ word '$w^x$' in $L^y$ is |

Table 12: The best template for each LLM respectively for zero-shot and few-shot prompting. For simplicity, we present only two in-context examples in each template in the few-shot setup.

| [5K Setup] | VecMap | ContrastiveBLI | mT5_small | mT5_base | mT5_large | mT5_xl | mT5_xxl | mT0_small | mT0_base | mT0_large | mT0_xl | mT0_xxl | XGLM_564M | XGLM_1.7B | XGLM_2.9B | XGLM_4.5B | XGLM_7.5B | mGPT | LLaMA_7B | LLaMA_13B |
|---|---|---|---|---|---|---|---|---|---|---|---|---|---|---|---|---|---|---|---|---|
| DE→FI | 33.59 | **44.65** | 7.82 | 11.79 | 25.56 | 31.72 | 35.63 | 0.63 | 4.49 | 3.91 | 15.81 | 27.18 | 20.55 | 31.61 | 35.89 | 32.97 | 39.96 | 19.67 | - | - |
| FI→DE | 38.73 | **47.03** | 10.98 | 17.76 | 32.79 | 37.89 | 41.99 | 2.52 | 7.09 | 6.15 | 19.81 | 27.75 | 16.5 | 31.48 | 40.25 | 40.04 | 40.15 | 24.96 | - | - |
| DE→FR | 50.44 | 55.56 | 14.08 | 23.53 | 39.12 | 47.1 | 51.07 | 2.45 | 10.28 | 11.95 | 28.95 | 36.83 | 19.72 | 34.95 | 43.09 | 49.5 | 48.57 | 37.04 | 56.13 | **59.68** |
| FR→DE | 47.75 | 53.29 | 12.78 | 21.11 | 37.35 | 41.54 | 44.49 | 2.17 | 7.71 | 7.4 | 21.68 | 32.95 | 14.38 | 27.42 | 40.82 | 45.89 | 42.89 | 31.45 | 51.58 | **53.75** |
| DE→HR | 32.08 | 42.41 | - | - | - | - | - | - | - | - | - | - | - | - | - | - | - | - | 39.65 | **44.08** |
| HR→DE | 37.24 | **48.29** | - | - | - | - | - | - | - | - | - | - | - | - | - | - | - | - | 43.08 | 46.71 |
| DE→IT | 50.55 | 54.77 | 12.26 | 22.27 | 36.67 | 43.45 | 48.15 | 1.41 | 6.26 | 7.2 | 24.36 | 35.47 | 18.41 | 32.81 | 43.77 | 48.51 | 45.23 | 39.96 | 53.63 | **57.75** |
| IT→DE | 47.29 | **53.8** | 11.78 | 19.28 | 34.88 | 38.81 | 43.67 | 2.58 | 7.49 | 7.03 | 20.31 | 28.94 | 14.06 | 26.1 | 36.74 | 41.55 | 38.4 | 29.15 | 47.86 | 51.52 |
| DE→RU | 34.38 | 46.79 | 8.61 | 22.33 | 35.16 | 41.84 | 43.24 | 0.0 | 0.37 | 0.26 | 17.63 | 31.82 | 19.77 | 34.85 | 41.99 | 44.08 | 44.03 | 32.19 | 47.57 | **49.45** |
| RU→DE | 43.32 | 49.71 | 10.11 | 21.9 | 37.3 | 41.17 | 44.21 | 1.52 | 4.61 | 8.38 | 21.11 | 29.39 | 14.56 | 24.83 | 40.54 | 43.84 | 41.44 | 31.22 | 48.35 | **50.39** |
| DE→TR | 27.18 | **38.86** | 7.3 | 12.31 | 25.2 | 29.53 | 36.41 | 1.46 | 4.54 | 4.49 | 15.18 | 21.44 | 13.15 | 19.25 | 28.59 | 30.41 | 31.92 | 22.9 | - | - |
| TR→DE | 29.93 | **40.95** | 8.31 | 13.26 | 28.75 | 32.48 | 38.18 | 1.6 | 6.28 | 5.38 | 17.2 | 24.87 | 11.08 | 18.8 | 26.09 | 32.11 | 28.86 | 24.07 | - | - |
| EN→DE | 51.0 | 57.75 | 16.8 | 30.25 | 45.9 | 49.35 | 53.1 | 3.3 | 8.55 | 8.45 | 25.5 | 39.35 | 21.15 | 38.6 | 52.5 | 54.95 | 50.3 | 38.05 | 62.3 | **63.45** |
| DE→EN | 55.24 | **58.95** | 19.51 | 30.26 | 42.62 | 49.5 | 53.94 | 0.37 | 9.65 | 12.0 | 29.94 | 38.81 | 29.11 | 42.31 | 47.94 | 51.64 | 51.17 | 40.58 | 57.28 | 58.89 |
| EN→FI | 37.75 | 47.15 | 9.6 | 18.75 | 34.25 | 40.8 | 44.65 | 1.3 | 4.35 | 3.4 | 19.15 | 30.25 | 25.4 | 39.9 | 48.1 | 41.7 | **48.65** | 21.3 | - | - |
| FI→EN | 43.51 | 50.55 | 10.98 | 21.97 | 36.63 | 43.51 | 48.82 | 0.16 | 4.83 | 6.36 | 21.39 | 34.05 | 28.64 | 44.19 | 50.24 | 46.14 | **53.18** | 27.59 | - | - |
| EN→FR | 63.1 | 67.2 | 23.95 | 39.5 | 55.6 | 62.45 | 66.8 | 5.0 | 15.5 | 16.95 | 38.75 | 50.95 | 32.35 | 55.95 | 63.8 | 67.8 | 66.75 | 50.25 | 72.95 | **76.25** |
| FR→EN | 62.75 | 65.49 | 27.16 | 42.42 | 50.75 | 57.84 | 61.41 | 0.47 | 14.85 | 17.95 | 37.4 | 47.28 | 40.87 | 54.22 | 57.37 | 60.99 | 60.01 | 50.44 | 64.82 | **67.3** |
| EN→HR | 34.05 | 47.2 | - | - | - | - | - | - | - | - | - | - | - | - | - | - | - | - | 48.15 | **54.7** |
| HR→EN | 39.08 | 49.08 | - | - | - | - | - | - | - | - | - | - | - | - | - | - | - | - | 48.13 | **51.03** |
| EN→IT | 60.4 | 65.6 | 21.45 | 38.3 | 54.75 | 61.75 | 63.85 | 2.3 | 5.9 | 6.35 | 34.05 | 48.0 | 29.7 | 51.1 | 60.65 | 64.55 | 62.9 | 52.4 | 71.4 | **74.0** |
| IT→EN | 62.17 | **65.27** | 20.26 | 35.35 | 47.03 | 54.32 | 59.43 | 0.31 | 10.8 | 14.42 | 33.49 | 43.0 | 34.68 | 48.17 | 53.23 | 57.73 | 57.11 | 48.63 | 63.05 | 64.44 |
| EN→RU | 39.65 | 50.5 | 11.85 | 29.6 | 45.6 | 52.8 | 53.7 | 0.2 | 0.35 | 0.25 | 22.0 | 38.5 | 30.95 | 46.4 | 54.0 | 57.5 | 55.05 | 42.05 | 58.3 | **62.3** |
| RU→EN | 49.35 | 54.16 | 12.31 | 29.39 | 43.16 | 51.86 | 54.58 | 0.16 | 4.3 | 11.05 | 27.76 | 36.83 | 30.12 | 43.79 | 52.28 | 53.95 | 53.64 | 40.96 | 56.1 | **58.77** |
| EN→TR | 32.05 | 44.75 | 10.55 | 20.65 | 35.45 | 40.15 | **45.05** | 2.65 | 4.2 | 3.8 | 18.6 | 27.05 | 16.9 | 29.35 | 38.4 | 40.6 | 40.85 | 31.05 | - | - |
| TR→EN | 39.24 | **44.78** | 9.42 | 20.77 | 32.43 | 39.51 | 44.36 | 0.16 | 3.83 | 7.61 | 18.0 | 30.4 | 19.81 | 33.07 | 39.24 | 39.83 | 42.17 | 29.71 | - | - |
| FI→FR | 38.26 | 45.24 | 8.57 | 18.5 | 35.47 | 42.35 | 46.51 | 1.63 | 6.2 | 5.83 | 24.54 | 32.05 | 20.91 | 37.94 | 46.66 | 45.45 | **48.77** | 28.69 | - | - |
| FR→FI | 34.3 | **43.2** | 6.93 | 12.42 | 27.32 | 34.2 | 37.61 | 0.83 | 3.16 | 2.64 | 15.99 | 25.3 | 20.49 | 29.07 | 39.16 | 35.28 | 42.52 | 19.66 | - | - |
| FI→HR | 31.58 | **38.31** | - | - | - | - | - | - | - | - | - | - | - | - | - | - | - | - | - | - |
| HR→FI | 31.72 | **39.56** | - | - | - | - | - | - | - | - | - | - | - | - | - | - | - | - | - | - |
| FI→IT | 37.99 | 46.3 | 10.3 | 17.5 | 32.16 | 39.83 | 44.51 | 1.52 | 6.78 | 5.25 | 21.97 | 30.06 | 21.86 | 36.57 | 44.04 | 44.09 | **47.08** | 27.75 | - | - |
| IT→FI | 34.32 | **43.57** | 7.7 | 11.73 | 25.22 | 32.09 | 35.09 | 0.88 | 4.55 | 4.29 | 16.18 | 23.41 | 17.26 | 29.35 | 36.43 | 31.47 | 40.67 | 18.35 | - | - |
| FI→RU | 34.16 | 40.99 | 6.36 | 16.34 | 30.53 | 38.05 | 41.04 | 0.05 | 0.53 | 0.68 | 15.55 | 27.01 | 19.23 | 36.1 | 42.3 | 39.99 | **44.82** | 24.38 | - | - |
| RU→FI | 33.53 | **40.91** | 4.56 | 13.51 | 26.72 | 34.31 | 35.57 | 0.37 | 1.94 | 2.78 | 14.14 | 24.1 | 16.29 | 27.45 | 34.36 | 34.47 | 40.81 | 18.54 | - | - |
| HR→FR | 40.24 | 49.29 | - | - | - | - | - | - | - | - | - | - | - | - | - | - | - | - | 46.66 | **50.34** |
| FR→HR | 33.21 | 44.08 | - | - | - | - | - | - | - | - | - | - | - | - | - | - | - | - | 40.3 | **46.51** |
| HR→IT | 40.24 | 48.97 | - | - | - | - | - | - | - | - | - | - | - | - | - | - | - | - | 46.87 | **50.6** |
| IT→HR | 34.32 | **44.75** | - | - | - | - | - | - | - | - | - | - | - | - | - | - | - | - | 38.5 | **44.75** |
| HR→RU | 37.98 | 46.4 | - | - | - | - | - | - | - | - | - | - | - | - | - | - | - | - | 43.29 | **46.87** |
| RU→HR | 39.5 | **45.47** | - | - | - | - | - | - | - | - | - | - | - | - | - | - | - | - | 40.07 | 45.42 |
| IT→FR | 65.89 | **67.86** | 16.43 | 31.63 | 49.87 | 56.59 | 59.84 | 2.53 | 12.56 | 13.49 | 34.83 | 44.03 | 23.67 | 41.76 | 53.44 | 58.81 | 55.71 | 49.04 | 63.51 | 66.77 |
| FR→IT | 64.72 | 67.2 | 17.64 | 33.83 | 51.37 | 57.68 | 60.42 | 1.5 | 5.33 | 5.59 | 32.64 | 44.85 | 23.07 | 41.54 | 57.37 | 60.27 | 59.18 | 54.58 | 65.08 | **68.03** |
| RU→FR | 47.51 | 52.7 | 11.26 | 24.2 | 41.12 | 48.82 | 51.44 | 1.31 | 6.23 | 9.17 | 26.51 | 35.78 | 21.74 | 35.57 | 46.36 | 52.38 | 51.13 | 39.55 | 55.32 | **58.04** |
| FR→RU | 38.23 | 48.06 | 8.64 | 23.33 | 38.18 | 44.65 | 47.54 | 0.0 | 0.47 | 0.26 | 19.61 | 33.73 | 25.92 | 39.89 | 46.82 | 50.08 | 49.77 | 39.63 | 51.22 | **54.16** |
| RU→IT | 46.78 | 51.96 | 11.79 | 26.14 | 41.59 | 47.77 | 49.29 | 0.89 | 4.45 | 7.18 | 24.83 | 34.36 | 23.47 | 32.58 | 46.88 | 50.13 | 50.6 | 41.28 | 53.54 | **57.83** |
| IT→RU | 38.71 | 47.49 | 8.22 | 22.02 | 36.9 | 43.82 | 46.41 | 0.1 | 0.36 | 0.26 | 18.66 | 30.96 | 22.58 | 36.9 | 43.26 | 47.39 | 47.75 | 35.45 | 46.61 | **51.89** |
| TR→FI | 28.59 | **34.77** | 5.54 | 8.47 | 20.18 | 27.1 | 29.77 | 0.43 | 3.3 | 2.08 | 14.11 | 18.85 | 13.47 | 22.79 | 26.94 | 25.03 | 32.8 | 16.61 | - | - |
| FI→TR | 29.8 | 35.68 | 7.67 | 12.19 | 22.86 | 31.0 | **36.21** | 1.68 | 5.52 | 3.84 | 14.66 | 20.13 | 14.61 | 23.44 | 31.21 | 30.22 | 35.73 | 20.18 | - | - |
| TR→FR | 36.58 | 43.88 | 8.95 | 17.25 | 31.79 | 39.99 | **44.62** | 1.38 | 4.95 | 5.86 | 22.63 | 29.5 | 15.55 | 25.83 | 33.76 | 37.7 | 38.82 | 29.55 | - | - |
| FR→TR | 31.76 | **42.06** | 8.48 | 14.38 | 30.42 | 34.97 | 40.61 | 2.28 | 4.81 | 3.36 | 16.81 | 23.49 | 13.4 | 22.76 | 31.71 | 33.94 | 34.71 | 27.78 | - | - |
| TR→HR | 25.99 | **36.32** | - | - | - | - | - | - | - | - | - | - | - | - | - | - | - | - | - | - |
| HR→TR | 27.35 | **37.09** | - | - | - | - | - | - | - | - | - | - | - | - | - | - | - | - | - | - |
| TR→IT | 34.24 | 42.17 | 7.35 | 15.28 | 30.46 | 38.13 | **42.55** | 1.22 | 5.32 | 4.58 | 19.97 | 28.12 | 14.64 | 24.17 | 35.46 | 36.85 | 37.01 | 31.63 | - | - |
| IT→TR | 30.7 | **40.62** | 8.11 | 14.21 | 26.25 | 31.94 | 37.26 | 2.12 | 5.37 | 3.36 | 16.12 | 22.02 | 12.61 | 21.71 | 31.01 | 32.09 | 34.63 | 25.06 | - | - |
| TR→RU | 26.2 | 36.16 | 4.47 | 13.53 | 29.39 | 35.57 | **38.02** | 0.0 | 0.27 | 0.43 | 13.63 | 25.03 | 13.26 | 27.32 | 34.45 | 34.93 | 36.47 | 27.21 | - | - |
| RU→TR | 27.08 | 35.78 | 5.97 | 14.35 | 27.08 | 31.69 | **36.09** | 1.47 | 3.25 | 4.24 | 14.51 | 19.91 | 11.58 | 19.75 | 30.23 | 32.63 | 35.73 | 24.62 | - | - |

Table 13: Full 5-shot BLI results (P@1×100%) on 56 XLING BLI directions with 5K seed translation pairs. Off-the-shelf LLMs are used without any fine-tuning. '-': a language in the pair is not supported by the LLM.

| [1K Setup] | VecMap | ContrastiveBLI | mT5$_{small}$ | mT5$_{base}$ | mT5$_{large}$ | mT5$_{xl}$ | mT5$_{xxl}$ | mT0$_{small}$ | mT0$_{base}$ | mT0$_{large}$ | mT0$_{xl}$ | mT0$_{xxl}$ | XGLM$_{564M}$ | XGLM$_{1.7B}$ | XGLM$_{2.9B}$ | XGLM$_{4.5B}$ | XGLM$_{7.5B}$ | mGPT | LLaMA$_{7B}$ | LLaMA$_{13B}$ |
|---|---|---|---|---|---|---|---|---|---|---|---|---|---|---|---|---|---|---|---|---|
| DE→FI | 28.33 | **43.77** | 6.0 | 10.8 | 21.91 | 28.95 | 34.12 | 0.37 | 3.65 | 3.6 | 13.72 | 23.53 | 18.78 | 30.41 | 35.32 | 31.56 | 35.84 | 18.31 | - | - |
| FI→DE | 35.0 | **44.93** | 8.57 | 13.87 | 26.38 | 33.58 | 36.94 | 0.53 | 5.83 | 3.84 | 16.5 | 25.38 | 16.97 | 28.74 | 38.47 | 35.94 | 37.2 | 21.18 | - | - |
| DE→FR | 49.03 | 54.04 | 10.12 | 21.02 | 35.32 | 43.56 | 48.46 | 1.2 | 9.6 | 8.87 | 24.93 | 34.38 | 20.4 | 36.46 | 43.77 | 48.93 | 47.0 | 37.66 | 54.77 | **56.76** |
| FR→DE | 44.34 | 48.16 | 9.57 | 17.64 | 32.33 | 36.37 | 41.75 | 0.72 | 6.47 | 5.79 | 18.37 | 30.21 | 13.3 | 25.35 | 38.23 | 42.47 | 39.68 | 31.4 | 49.15 | **51.94** |
| DE→HR | 27.39 | 40.48 | - | - | - | - | - | - | - | - | - | - | - | - | - | - | - | - | 36.57 | **41.78** |
| HR→DE | 32.82 | **44.35** | - | - | - | - | - | - | - | - | - | - | - | - | - | - | - | - | 39.61 | 43.19 |
| DE→IT | 48.72 | 52.53 | 8.92 | 16.95 | 33.23 | 41.16 | 44.91 | 0.89 | 5.74 | 6.21 | 24.26 | 33.91 | 19.67 | 33.91 | 42.46 | 45.44 | 44.03 | 37.4 | 51.38 | **55.92** |
| IT→DE | 44.39 | 49.66 | 9.25 | 16.38 | 29.15 | 33.23 | 40.93 | 0.83 | 5.58 | 5.79 | 19.79 | 27.65 | 13.07 | 26.3 | 36.02 | 39.84 | 37.62 | 28.89 | 45.06 | **50.18** |
| DE→RU | 25.46 | 42.83 | 6.68 | 20.08 | 32.45 | 39.02 | 39.75 | 0.05 | 0.21 | 0.37 | 15.6 | 28.38 | 19.2 | 33.39 | 40.01 | 43.19 | 40.38 | 30.36 | 45.49 | **47.63** |
| RU→DE | 39.08 | 46.99 | 7.18 | 20.43 | 34.15 | 37.45 | 41.8 | 0.26 | 3.09 | 6.13 | 19.02 | 28.55 | 14.56 | 24.46 | 39.65 | 41.49 | 39.03 | 32.11 | 44.74 | **48.45** |
| DE→TR | 23.37 | **34.85** | 4.75 | 9.08 | 21.28 | 27.8 | 32.81 | 0.83 | 3.65 | 3.44 | 13.2 | 17.84 | 12.0 | 18.78 | 25.09 | 27.44 | 28.9 | 22.33 | - | - |
| TR→DE | 26.57 | **37.11** | 5.96 | 9.32 | 23.0 | 28.43 | 32.53 | 0.37 | 4.58 | 3.78 | 13.74 | 22.2 | 10.17 | 17.52 | 25.03 | 28.65 | 27.16 | 22.26 | - | - |
| EN→DE | 48.2 | 50.85 | 15.05 | 27.1 | 40.85 | 46.75 | 50.0 | 1.0 | 7.85 | 7.7 | 21.8 | 35.9 | 20.85 | 35.15 | 51.5 | 52.45 | 48.95 | 35.2 | 59.0 | **62.85** |
| DE→EN | 54.56 | **57.75** | 16.9 | 28.22 | 37.19 | 45.85 | 50.97 | 0.21 | 9.44 | 9.44 | 28.12 | 37.61 | 28.38 | 40.74 | 45.91 | 50.03 | 48.41 | 38.5 | 54.46 | 54.98 |
| EN→FI | 27.95 | 45.0 | 6.9 | 14.6 | 29.7 | 36.9 | 40.45 | 0.45 | 3.35 | 2.9 | 14.55 | 25.0 | 25.3 | 37.65 | 46.5 | 39.55 | **47.3** | 18.4 | - | - |
| FI→EN | 41.15 | 48.77 | 8.88 | 16.87 | 28.43 | 36.0 | 42.83 | 0.0 | 4.1 | 2.42 | 17.24 | 30.27 | 28.85 | 41.99 | 46.77 | 43.72 | **49.61** | 23.91 | - | - |
| EN→FR | 60.0 | 62.5 | 21.25 | 37.35 | 51.65 | 59.6 | 63.6 | 2.55 | 14.2 | 13.0 | 33.45 | 47.3 | 33.05 | 53.45 | 62.2 | 66.65 | 65.8 | 50.1 | 71.2 | **76.0** |
| FR→EN | 61.41 | **64.05** | 22.04 | 37.09 | 43.77 | 53.08 | 56.54 | 0.52 | 14.07 | 10.97 | 33.83 | 45.42 | 40.25 | 51.22 | 55.04 | 57.63 | 54.99 | 48.42 | 61.1 | 61.56 |
| EN→HR | 24.95 | 42.35 | - | - | - | - | - | - | - | - | - | - | - | - | - | - | - | - | 45.95 | **53.4** |
| HR→EN | 37.45 | **47.55** | - | - | - | - | - | - | - | - | - | - | - | - | - | - | - | - | 42.08 | 45.61 |
| EN→IT | 57.55 | 61.05 | 17.4 | 33.75 | 49.35 | 58.15 | 59.65 | 0.9 | 5.0 | 5.05 | 30.55 | 42.9 | 29.95 | 50.65 | 57.85 | 63.4 | 59.8 | 51.2 | 69.35 | **73.1** |
| IT→EN | 60.78 | **63.67** | 16.59 | 31.99 | 41.24 | 51.52 | 54.73 | 0.21 | 10.03 | 7.86 | 31.16 | 40.05 | 34.63 | 46.67 | 49.51 | 54.94 | 52.14 | 46.87 | 56.95 | 59.22 |
| EN→RU | 25.05 | 46.05 | 11.3 | 27.2 | 42.05 | 49.25 | 51.5 | 0.05 | 0.15 | 0.15 | 18.4 | 33.95 | 30.55 | 44.1 | 52.75 | 54.7 | 53.25 | 41.7 | 56.65 | **61.15** |
| RU→EN | 46.2 | **53.17** | 8.96 | 25.3 | 38.55 | 45.68 | 49.19 | 0.0 | 3.25 | 5.03 | 26.45 | 35.31 | 30.59 | 42.48 | 49.66 | 51.44 | 49.4 | 38.92 | 51.65 | 52.49 |
| EN→TR | 26.7 | 41.05 | 6.0 | 15.25 | 29.3 | 35.9 | **42.35** | 0.35 | 3.1 | 2.8 | 13.15 | 20.75 | 15.25 | 24.6 | 33.75 | 34.6 | 37.85 | 27.75 | - | - |
| TR→EN | 37.17 | **43.24** | 8.09 | 15.02 | 25.99 | 34.56 | 40.42 | 0.16 | 2.61 | 2.24 | 15.65 | 28.17 | 18.58 | 28.65 | 34.98 | 36.53 | 37.27 | 28.06 | - | - |
| FI→FR | 34.79 | 43.3 | 7.3 | 13.93 | 27.96 | 38.05 | 41.67 | 1.0 | 5.1 | 3.0 | 21.28 | 29.53 | 22.75 | 37.83 | 44.98 | 42.98 | **47.92** | 32.11 | - | - |
| FR→FI | 23.95 | **40.56** | 3.67 | 9.78 | 24.42 | 29.33 | 33.89 | 0.26 | 2.79 | 2.07 | 12.62 | 22.87 | 20.43 | 31.25 | 38.49 | 32.44 | 39.11 | 17.54 | - | - |
| FI→HR | 29.9 | **34.26** | - | - | - | - | - | - | - | - | - | - | - | - | - | - | - | - | - | - |
| HR→FI | 27.62 | **36.14** | - | - | - | - | - | - | - | - | - | - | - | - | - | - | - | - | - | - |
| FI→IT | 34.68 | 42.88 | 6.67 | 14.45 | 26.06 | 35.21 | 37.99 | 0.58 | 6.04 | 2.79 | 18.76 | 26.75 | 23.12 | 37.41 | 42.2 | 41.36 | **45.19** | 24.12 | - | - |
| IT→FI | 26.1 | **41.65** | 4.6 | 8.58 | 21.86 | 28.79 | 32.71 | 0.47 | 4.5 | 3.67 | 13.64 | 21.19 | 16.69 | 29.92 | 34.99 | 30.85 | 37.0 | 17.21 | - | - |
| FI→RU | 30.27 | 37.15 | 4.36 | 12.93 | 25.07 | 33.74 | 37.52 | 0.0 | 0.42 | 0.21 | 12.66 | 24.65 | 17.71 | 34.89 | 40.41 | 38.36 | **42.98** | 22.39 | - | - |
| RU→FI | 33.11 | 37.35 | 2.83 | 9.8 | 24.1 | 30.07 | 33.05 | 0.05 | 1.57 | 1.83 | 13.36 | 21.16 | 16.61 | 27.24 | 34.0 | 33.21 | **38.08** | 17.18 | - | - |
| HR→FR | 39.14 | 45.71 | - | - | - | - | - | - | - | - | - | - | - | - | - | - | - | - | 44.29 | **47.66** |
| FR→HR | 27.52 | 39.68 | - | - | - | - | - | - | - | - | - | - | - | - | - | - | - | - | 38.33 | **43.82** |
| HR→IT | 38.77 | 46.19 | - | - | - | - | - | - | - | - | - | - | - | - | - | - | - | - | 44.56 | **46.92** |
| IT→HR | 28.68 | 41.29 | - | - | - | - | - | - | - | - | - | - | - | - | - | - | - | - | 36.02 | **43.15** |
| HR→RU | 36.09 | 42.14 | - | - | - | - | - | - | - | - | - | - | - | - | - | - | - | - | 40.08 | **44.45** |
| RU→HR | 38.08 | 41.17 | - | - | - | - | - | - | - | - | - | - | - | - | - | - | - | - | 39.65 | **44.58** |
| IT→FR | 65.06 | **66.77** | 15.3 | 28.27 | 45.53 | 53.8 | 56.64 | 1.29 | 11.99 | 9.46 | 32.14 | 40.16 | 22.89 | 42.27 | 54.06 | 59.53 | 55.97 | 48.06 | 60.72 | 64.81 |
| FR→IT | 63.58 | 65.49 | 13.45 | 26.64 | 45.63 | 52.97 | 56.03 | 0.72 | 5.38 | 4.91 | 30.99 | 42.47 | 22.56 | 44.44 | 56.8 | 58.04 | 57.37 | 53.91 | 63.89 | **66.68** |
| RU→FR | 45.42 | 51.28 | 7.81 | 22.26 | 38.24 | 45.89 | 49.35 | 0.84 | 5.87 | 4.66 | 25.46 | 33.68 | 23.21 | 36.14 | 47.15 | 51.86 | 49.4 | 39.76 | 53.06 | **55.47** |
| FR→RU | 24.57 | 43.61 | 7.09 | 20.33 | 34.76 | 41.96 | 43.51 | 0.05 | 0.16 | 0.21 | 16.55 | 30.01 | 23.49 | 38.08 | 43.82 | 46.2 | 46.3 | 37.56 | 48.16 | **53.23** |
| RU→IT | 43.74 | 49.24 | 7.96 | 22.11 | 37.51 | 43.27 | 46.41 | 0.31 | 3.2 | 5.19 | 23.99 | 33.32 | 22.16 | 35.31 | 45.99 | 49.87 | 47.88 | 39.03 | 51.65 | **54.58** |
| IT→RU | 26.1 | 43.57 | 6.51 | 18.14 | 33.8 | 40.36 | 41.65 | 0.05 | 0.31 | 0.21 | 15.19 | 27.24 | 20.88 | 36.64 | 40.98 | 45.58 | 44.34 | 35.61 | 43.98 | **50.03** |
| TR→FI | 24.76 | 32.96 | 3.94 | 5.27 | 16.51 | 22.95 | 25.61 | 0.27 | 3.04 | 1.65 | 11.5 | 17.84 | 12.73 | 22.42 | 26.14 | 23.16 | 29.45 | 14.27 | - | - |
| FI→TR | 25.8 | 30.64 | 4.52 | 8.36 | 18.13 | 23.49 | 30.06 | 0.21 | 3.52 | 1.94 | 11.19 | 18.08 | 13.82 | 21.65 | 28.74 | 26.17 | **32.74** | 17.34 | - | - |
| TR→FR | 32.85 | **41.59** | 6.28 | 13.84 | 26.57 | 36.0 | 41.37 | 0.59 | 3.78 | 2.5 | 20.34 | 28.12 | 14.91 | 26.14 | 33.81 | 36.42 | 37.49 | 30.14 | - | - |
| FR→TR | 25.19 | **38.44** | 4.81 | 10.61 | 24.88 | 28.5 | 34.87 | 0.52 | 2.95 | 2.33 | 12.52 | 18.21 | 13.24 | 21.31 | 29.49 | 29.95 | 31.66 | 26.49 | - | - |
| TR→HR | 20.5 | **32.16** | - | - | - | - | - | - | - | - | - | - | - | - | - | - | - | - | - | - |
| HR→TR | 20.67 | **33.04** | - | - | - | - | - | - | - | - | - | - | - | - | - | - | - | - | - | - |
| TR→IT | 31.42 | **39.19** | 5.43 | 10.81 | 26.2 | 33.92 | 38.39 | 0.53 | 3.99 | 2.93 | 18.0 | 25.67 | 12.78 | 24.92 | 33.28 | 35.09 | 35.57 | 31.79 | - | - |
| IT→TR | 25.06 | **37.93** | 3.98 | 9.2 | 22.12 | 27.03 | 33.59 | 0.36 | 3.15 | 2.95 | 13.33 | 17.73 | 12.14 | 19.38 | 25.68 | 28.53 | 29.82 | 22.95 | - | - |
| TR→RU | 15.55 | 31.95 | 3.51 | 11.87 | 23.75 | 32.37 | **35.52** | 0.0 | 0.27 | 0.1 | 12.73 | 23.06 | 12.41 | 25.61 | 31.2 | 33.87 | 34.19 | 25.56 | - | - |
| RU→TR | 18.6 | 33.05 | 3.14 | 10.53 | 23.78 | 28.34 | **34.15** | 0.1 | 1.57 | 2.1 | 10.63 | 16.61 | 10.42 | 18.07 | 27.4 | 30.23 | 32.16 | 22.21 | - | - |

Table 14: Full 5-shot BLI results (P@1×100%) on 56 XLING BLI directions with 1K seed translation pairs. Off-the-shelf LLMs are used without any fine-tuning. '-': a language in the pair is not supported by the LLM.

| [1K Setup] | VecMap | ContrastiveBLI | mT5$_{small}$ | mT5$_{base}$ | mT5$_{large}$ | mT5$_{xl}$ | mT5$_{xxl}$ | mT0$_{small}$ | mT0$_{base}$ | mT0$_{large}$ | mT0$_{xl}$ | mT0$_{xxl}$ | XGLM$_{564M}$ | XGLM$_{1.7B}$ | XGLM$_{2.9B}$ | XGLM$_{4.5B}$ | XGLM$_{7.5B}$ | mGPT | LLaMA$_{7B}$ | LLaMA$_{13B}$ |
|---|---|---|---|---|---|---|---|---|---|---|---|---|---|---|---|---|---|---|---|---|
| BG→CA | 39.66 | **43.93** | 4.56 | 14.25 | 25.88 | 34.29 | 38.67 | 0.18 | 3.1 | 6.19 | 23.19 | 31.72 | 15.83 | 24.01 | 35.69 | 33.06 | 40.19 | - | 37.85 | 41.71 |
| CA→BG | 33.54 | 40.06 | 3.74 | 11.27 | 24.48 | 30.54 | 36.2 | 0.11 | 0.34 | 0.11 | 10.03 | 22.72 | 15.86 | 29.8 | 37.45 | 33.09 | 40.23 | - | 36.66 | **41.53** |
| BG→HU | 38.77 | **44.62** | 2.2 | 9.66 | 22.34 | 31.48 | 36.17 | 0.06 | 1.56 | 4.98 | 14.87 | 25.46 | - | - | - | - | - | 23.61 | 31.54 | 36.57 |
| HU→BG | 36.52 | 43.03 | 3.17 | 11.29 | 22.64 | 34.27 | 36.98 | 0.0 | 0.23 | 0.17 | 9.97 | 24.02 | - | - | - | - | - | 28.17 | 37.5 | **43.2** |
| CA→HU | 35.47 | **41.44** | 4.64 | 10.33 | 19.67 | 27.18 | 32.43 | 0.5 | 3.92 | 4.92 | 12.98 | 22.21 | - | - | - | - | - | - | 30.39 | 35.3 |
| HU→CA | 39.88 | **47.14** | 6.38 | 12.65 | 20.19 | 34.98 | 37.68 | 0.61 | 5.89 | 6.88 | 19.86 | 29.59 | - | - | - | - | - | - | 39.55 | 46.04 |
| Avg. | 37.31 | **43.37** | 4.11 | 11.58 | 22.53 | 32.12 | 36.36 | 0.24 | 2.51 | 3.88 | 15.15 | 25.95 | - | - | - | - | - | - | 35.58 | 40.72 |

Table 15: 5-shot BLI results (P@1×100%) on PanLex-BLI with 1K seed translation pairs. Off-the-shelf LLMs are used without any fine-tuning. '-': a language in the pair is not supported by the LLM.

Table 16:

| [Zero-Shot] | VecMap | mT5$_{small}$ | mT5$_{base}$ | mT5$_{large}$ | mT5$_{xl}$ | mT5$_{xxl}$ | mT0$_{small}$ | mT0$_{base}$ | mT0$_{large}$ | mT0$_{xl}$ | mT0$_{xxl}$ | XGLM$_{564M}$ | XGLM$_{1.7B}$ | XGLM$_{2.9B}$ | XGLM$_{4.5B}$ | XGLM$_{7.5B}$ | mGPT | LLaMA$_{7B}$ | LLaMA$_{13B}$ |
|---|---|---|---|---|---|---|---|---|---|---|---|---|---|---|---|---|---|---|---|
| DE→FI | 28.27 | 6.31 | 7.25 | 12.99 | 23.37 | 29.63 | 4.69 | 4.02 | 5.63 | 14.24 | 22.07 | 9.39 | 19.09 | 26.19 | 25.67 | **30.1** | 11.89 | - | - |
| FI→DE | 34.84 | 7.46 | 7.83 | 17.29 | 22.12 | 29.53 | 6.15 | 5.52 | 7.36 | 10.04 | 23.75 | 8.99 | 24.49 | 31.79 | 29.48 | **35.26** | 13.14 | - | - |
| DE→FR | **49.19** | 9.96 | 12.73 | 25.35 | 34.69 | 38.03 | 10.02 | 6.31 | 19.72 | 19.77 | 30.26 | 10.75 | 24.2 | 29.73 | 39.44 | 37.3 | 27.18 | 42.41 | 47.37 |
| FR→DE | 44.44 | 10.09 | 11.23 | 26.28 | 34.14 | 41.33 | 8.85 | 7.97 | 9.0 | 18.52 | 31.82 | 11.43 | 26.28 | 33.11 | 40.82 | 37.66 | 22.97 | 43.15 | **48.11** |
| DE→HR | 27.49 | - | - | - | - | - | - | - | - | - | - | - | - | - | - | - | - | **29.06** | 28.06 |
| HR→DE | 32.72 | - | - | - | - | - | - | - | - | - | - | - | - | - | - | - | - | 31.61 | **35.72** |
| DE→IT | **48.41** | 8.09 | 12.21 | 24.47 | 33.44 | 37.09 | 7.62 | 8.24 | 9.75 | 22.12 | 31.25 | 11.42 | 21.96 | 30.36 | 38.29 | 37.09 | 28.01 | 42.83 | 45.96 |
| IT→DE | 43.93 | 8.27 | 9.82 | 21.65 | 28.63 | 36.43 | 7.29 | 6.46 | 8.79 | 17.67 | 27.29 | 9.72 | 23.2 | 29.04 | 34.63 | 30.23 | 22.53 | 38.76 | **44.39** |
| DE→RU | 25.67 | 0.42 | 0.47 | 20.24 | 28.33 | 28.69 | 0.42 | 1.72 | 0.26 | 14.19 | 20.76 | 1.04 | 19.09 | 20.61 | **33.85** | 30.2 | 13.51 | 30.46 | 32.86 |
| RU→DE | 38.97 | 1.78 | 8.17 | 24.93 | 33.47 | 41.33 | 3.72 | 4.92 | 7.75 | 16.87 | 28.6 | 6.71 | 15.87 | 32.11 | 37.82 | 35.31 | 15.98 | **43.48** | 43.11 |
| DE→TR | 23.79 | 5.01 | 6.89 | 15.28 | 18.62 | **26.4** | 4.43 | 4.96 | 5.37 | 11.53 | 18.1 | 4.28 | 10.12 | 16.69 | 19.51 | 19.82 | 11.48 | - | - |
| TR→DE | 26.46 | 7.08 | 6.98 | 13.74 | 18.58 | 24.81 | 7.24 | 6.6 | 6.92 | 12.0 | 19.33 | 4.69 | 14.59 | 17.73 | 24.17 | 21.83 | 14.38 | - | - |
| EN→DE | 48.45 | 11.55 | 11.85 | 35.35 | 42.5 | 50.5 | 10.6 | 9.4 | 10.55 | 24.0 | 39.3 | 15.55 | 32.05 | 43.7 | 49.75 | 46.4 | 23.95 | 52.05 | **54.5** |
| DE→EN | 54.51 | 15.6 | 17.37 | 23.47 | 36.25 | 39.23 | 14.14 | 11.06 | 24.62 | 29.16 | 34.95 | 15.86 | 26.71 | 36.31 | 41.89 | 31.3 | 21.91 | 42.57 | 47.0 |
| EN→FI | 28.15 | 5.25 | 5.85 | 17.35 | 29.1 | 37.15 | 4.2 | 4.25 | 5.25 | 15.65 | 27.45 | 9.3 | 21.65 | 34.35 | 31.1 | **37.7** | 12.1 | - | - |
| FI→EN | **41.04** | 8.04 | 8.67 | 14.98 | 23.7 | 32.0 | 5.94 | 5.2 | 13.45 | 19.02 | 27.27 | 8.72 | 17.03 | 39.78 | 29.48 | 39.57 | 14.14 | - | - |
| EN→FR | 60.1 | 16.95 | 19.95 | 43.65 | 51.95 | 56.25 | 16.9 | 11.55 | 29.7 | 22.7 | 32.25 | 19.5 | 38.7 | 50.95 | 56.25 | 52.15 | 35.9 | 57.6 | **62.7** |
| FR→EN | 61.51 | 23.64 | 25.25 | 31.97 | 44.9 | 47.44 | 21.68 | 17.74 | 35.8 | 37.97 | 44.49 | 22.56 | 38.7 | 47.08 | 50.85 | 39.32 | 38.13 | 54.53 | 57.32 |
| EN→HR | 24.1 | - | - | - | - | - | - | - | - | - | - | - | - | - | - | - | - | **35.85** | 28.6 |
| HR→EN | 36.3 | - | - | - | - | - | - | - | - | - | - | - | - | - | - | - | - | 31.83 | **37.35** |
| EN→IT | 57.4 | 8.05 | 13.25 | 38.4 | 46.95 | 54.2 | 7.25 | 9.55 | 11.05 | 31.1 | 44.3 | 17.8 | 35.4 | 47.85 | 53.95 | 50.55 | 37.85 | 58.9 | **60.4** |
| IT→EN | **60.78** | 14.37 | 17.57 | 25.48 | 37.47 | 42.58 | 16.33 | 12.61 | 27.91 | 33.02 | 37.88 | 19.12 | 31.52 | 39.64 | 43.82 | 35.09 | 26.25 | 47.39 | 54.88 |
| EN→RU | 25.1 | 0.35 | 0.4 | 31.85 | 38.6 | 42.8 | 0.4 | 1.95 | 0.25 | 19.0 | 30.05 | 1.6 | 24.65 | 37.35 | **44.1** | 42.3 | 16.7 | 42.15 | 40.15 |
| RU→EN | 46.41 | 2.25 | 12.99 | 23.83 | 36.88 | 43.37 | 4.71 | 5.97 | 23.57 | 27.29 | 37.77 | 8.85 | 17.65 | 40.86 | 42.43 | 39.65 | 16.82 | 46.2 | **50.18** |
| EN→TR | 26.5 | 4.05 | 5.3 | 20.0 | 26.95 | **35.05** | 3.9 | 4.9 | 4.65 | 13.05 | 24.9 | 5.2 | 9.4 | 21.8 | 22.35 | 27.1 | 10.55 | - | - |
| TR→EN | **36.9** | 6.82 | 7.45 | 13.26 | 22.47 | 27.9 | 5.96 | 3.73 | 12.78 | 17.25 | 21.88 | 5.91 | 15.12 | 24.81 | 26.46 | 25.99 | 7.14 | - | - |
| FI→FR | 35.21 | 5.25 | 7.04 | 15.34 | 24.38 | 27.69 | 5.25 | 2.89 | 10.35 | 13.82 | 19.81 | 8.25 | 24.44 | 33.74 | 30.16 | 35.0 | 16.13 | - | - |
| FR→FI | 22.56 | 4.76 | 5.43 | 11.9 | 22.81 | 29.38 | 3.1 | 3.26 | 4.86 | 14.23 | 22.56 | 7.24 | 19.61 | 27.57 | 26.54 | **30.78** | 11.48 | - | - |
| FI→HR | **28.8** | - | - | - | - | - | - | - | - | - | - | - | - | - | - | - | - | - | - |
| HR→FI | **27.72** | - | - | - | - | - | - | - | - | - | - | - | - | - | - | - | - | - | - |
| FI→IT | 34.79 | 6.73 | 6.88 | 13.61 | 20.23 | 26.85 | 6.2 | 6.57 | 6.94 | 15.03 | 23.54 | 9.67 | 24.28 | 33.79 | 28.9 | **37.05** | 16.45 | - | - |
| IT→FI | 25.99 | 5.89 | 6.61 | 8.32 | 18.97 | **26.46** | 4.34 | 3.77 | 5.58 | 11.63 | 18.86 | 6.82 | 18.6 | 24.6 | 23.57 | 25.53 | 12.04 | - | - |
| FI→RU | **31.11** | 0.32 | 0.42 | 14.87 | 20.6 | 24.8 | 0.37 | 1.21 | 0.26 | 10.61 | 20.34 | 0.53 | 22.23 | 23.44 | 26.33 | 26.33 | 8.41 | - | - |
| RU→FI | **32.74** | 1.0 | 4.87 | 12.94 | 21.06 | 31.74 | 0.84 | 1.68 | 3.2 | 12.0 | 20.53 | 4.98 | 9.64 | 16.34 | 24.88 | 29.65 | 7.39 | - | - |
| HR→FR | **38.82** | - | - | - | - | - | - | - | - | - | - | - | - | - | - | - | - | 31.51 | 36.24 |
| FR→HR | 26.95 | - | - | - | - | - | - | - | - | - | - | - | - | - | - | - | - | **31.61** | 31.09 |
| HR→IT | **39.03** | - | - | - | - | - | - | - | - | - | - | - | - | - | - | - | - | 34.46 | 38.51 |
| IT→HR | **28.84** | - | - | - | - | - | - | - | - | - | - | - | - | - | - | - | - | 28.58 | **28.84** |
| HR→RU | **36.24** | - | - | - | - | - | - | - | - | - | - | - | - | - | - | - | - | 30.51 | 31.77 |
| RU→HR | **38.71** | - | - | - | - | - | - | - | - | - | - | - | - | - | - | - | - | 34.31 | 10.9 |
| IT→FR | **65.22** | 9.46 | 15.66 | 29.51 | 41.71 | 47.86 | 13.28 | 9.25 | 26.46 | 25.27 | 35.97 | 13.44 | 32.97 | 38.6 | 46.87 | 41.91 | 42.27 | 51.47 | 57.42 |
| FR→IT | 63.42 | 8.28 | 14.9 | 35.33 | 45.42 | 49.92 | 7.97 | 9.73 | 12.83 | 32.64 | 41.85 | 16.09 | 33.73 | 46.2 | 52.46 | 49.46 | 44.75 | 57.17 | 60.01 |
| RU→FR | 45.31 | 1.73 | 10.74 | 28.03 | 37.98 | 42.01 | 4.24 | 3.82 | 20.38 | 21.32 | 30.17 | 6.65 | 16.19 | 31.85 | 40.6 | 38.92 | 24.62 | 43.64 | **47.98** |
| FR→RU | 23.59 | 0.36 | 0.41 | 24.16 | 31.14 | 30.78 | 0.31 | 1.66 | 0.26 | 15.73 | 24.47 | 2.33 | 22.66 | 28.04 | 36.83 | 33.32 | 17.07 | 35.8 | **38.59** |
| RU→IT | 43.84 | 1.52 | 11.31 | 25.25 | 36.09 | 41.49 | 3.35 | 5.61 | 9.27 | 21.79 | 32.22 | 8.96 | 14.09 | 34.36 | 42.54 | 38.87 | 28.55 | **47.2** | 45.99 |
| IT→RU | 25.94 | 0.41 | 0.41 | 19.84 | 26.46 | 28.48 | 0.41 | 1.96 | 0.26 | 13.18 | 22.74 | 1.5 | 20.83 | 20.52 | 31.01 | 26.72 | 18.55 | 31.52 | **35.45** |
| TR→FI | **19.91** | 4.47 | 4.74 | 6.44 | 13.68 | 18.69 | 3.35 | 3.73 | 4.42 | 9.53 | 15.02 | 3.51 | 13.37 | 16.24 | 18.64 | 17.31 | 9.32 | - | - |
| FI→TR | **20.97** | 4.26 | 5.52 | 9.56 | 13.87 | 19.76 | 3.21 | 4.15 | 5.1 | 8.57 | 16.97 | 4.68 | 12.45 | 18.97 | 17.13 | 20.55 | 6.36 | - | - |
| TR→FR | **32.96** | 5.11 | 6.44 | 14.7 | 22.36 | 23.8 | 4.47 | 3.51 | 11.82 | 12.14 | 17.31 | 3.78 | 15.02 | 18.21 | 27.21 | 24.65 | 20.29 | - | - |
| FR→TR | 25.04 | 3.83 | 5.59 | 15.57 | 20.95 | **27.47** | 3.21 | 4.29 | 4.66 | 10.5 | 20.18 | 3.93 | 9.36 | 17.9 | 20.28 | 20.28 | 11.38 | - | - |
| TR→HR | **21.14** | - | - | - | - | - | - | - | - | - | - | - | - | - | - | - | - | - | - |
| HR→TR | **20.62** | - | - | - | - | - | - | - | - | - | - | - | - | - | - | - | - | - | - |
| TR→IT | **31.84** | 5.43 | 6.39 | 12.03 | 20.29 | 24.81 | 5.17 | 5.59 | 7.77 | 14.54 | 20.39 | 4.9 | 14.43 | 20.39 | 26.2 | 25.35 | 21.88 | - | - |
| IT→TR | **25.06** | 4.5 | 5.79 | 13.64 | 17.67 | 24.91 | 3.41 | 4.13 | 4.96 | 10.23 | 18.04 | 4.29 | 10.59 | 16.59 | 18.29 | 18.6 | 13.49 | - | - |
| TR→RU | 12.09 | 0.43 | 0.48 | 11.4 | 19.33 | 20.98 | 0.48 | 1.28 | 0.27 | 7.67 | 16.24 | 0.16 | 14.0 | 11.08 | **22.95** | 14.32 | 11.61 | - | - |
| RU→TR | 15.61 | 1.0 | 4.71 | 15.87 | 20.95 | **27.97** | 1.41 | 2.99 | 3.46 | 10.27 | 16.5 | 2.2 | 4.19 | 16.92 | 19.02 | 20.32 | 8.75 | - | - |

Table 16: Full zero-shot BLI results (P@1×100%) on 56 XLING BLI directions without any seed translation pair (our zero-shot setup is also known as the unsupervised BLI setup for CLWE-based approaches). Off-the-shelf LLMs are used without any fine-tuning. '-': a language in the pair is not supported by the LLM.

Table 17:

| [Zero-Shot] | VecMap | mT5$_{small}$ | mT5$_{base}$ | mT5$_{large}$ | mT5$_{xl}$ | mT5$_{xxl}$ | mT0$_{small}$ | mT0$_{base}$ | mT0$_{large}$ | mT0$_{xl}$ | mT0$_{xxl}$ | XGLM$_{564M}$ | XGLM$_{1.7B}$ | XGLM$_{2.9B}$ | XGLM$_{4.5B}$ | XGLM$_{7.5B}$ | mGPT | LLaMA$_{7B}$ | LLaMA$_{13B}$ |
|---|---|---|---|---|---|---|---|---|---|---|---|---|---|---|---|---|---|---|---|
| BG→CA | **39.6** | 0.82 | 4.79 | 17.23 | 25.06 | 28.04 | 2.16 | 1.4 | 15.19 | 15.6 | 28.86 | 5.55 | 7.24 | 22.72 | 25.29 | 28.5 | - | 32.83 | 32.77 |
| CA→BG | **33.6** | 0.4 | 0.51 | 2.66 | 15.64 | 21.47 | 0.4 | 0.74 | 0.45 | 6.86 | 16.83 | 0.4 | 16.26 | 21.93 | 19.55 | 20.17 | - | 26.35 | 27.03 |
| BG→HU | **39.24** | 0.93 | 5.32 | 14.7 | 21.3 | 27.26 | 1.45 | 4.63 | 7.58 | 12.04 | 24.07 | - | - | - | - | 37.3 | 7.23 | 24.19 | 23.61 |
| HU→BG | **36.46** | 0.35 | 0.29 | 0.92 | 17.8 | 22.47 | 0.23 | 0.58 | 0.23 | 5.93 | 16.94 | - | - | - | - | - | 9.85 | 28.34 | 26.5 |
| CA→HU | **34.09** | 6.85 | 8.34 | 13.31 | 21.55 | 24.59 | 6.8 | 8.34 | 7.57 | 12.54 | 22.93 | - | - | - | - | - | - | 23.7 | 24.53 |
| HU→CA | 37.79 | 6.6 | 6.27 | 16.72 | 24.2 | 25.47 | 8.64 | 4.84 | 14.03 | 14.8 | 24.48 | - | - | - | - | - | - | 32.62 | **38.17** |
| Avg. | **36.8** | 2.66 | 4.25 | 10.92 | 20.92 | 24.88 | 3.28 | 3.42 | 7.51 | 11.3 | 22.35 | - | - | - | - | - | - | 28.0 | 28.77 |

Table 17: Zero-shot BLI results (P@1×100%) on PanLex-BLI without any seed translation pair (our zero-shot setup is also known as the unsupervised BLI setup for CLWE-based approaches). Off-the-shelf LLMs are used without any fine-tuning. '-': a language in the pair is not supported by the LLM.

Table 18:

| | mT5$_{small}$ | mT5$_{base}$ | mT5$_{large}$ | mT5$_{xl}$ | mT5$_{xxl}$ | mT0$_{small}$ | mT0$_{base}$ | mT0$_{large}$ | mT0$_{xl}$ | mT0$_{xxl}$ | XGLM$_{564M}$ | XGLM$_{1.7B}$ | XGLM$_{2.9B}$ | XGLM$_{4.5B}$ | XGLM$_{7.5B}$ | mGPT | LLaMA$_{7B}$ | LLaMA$_{13B}$ |
|---|---|---|---|---|---|---|---|---|---|---|---|---|---|---|---|---|---|---|
| NN (5K) | 14.84 | 28.33 | 43.25 | 49.66 | 52.83 | 1.43 | 6.8 | 8.28 | 27.0 | 38.09 | 24.51 | 39.49 | 49.14 | 53.08 | 51.58 | 41.2 | 57.33 | 60.23 |
| NN (1K) | 11.97 | 24.91 | 38.84 | 45.95 | 49.37 | 0.63 | 6.08 | 5.85 | 24.5 | 35.42 | 24.18 | 38.86 | 47.64 | 51.18 | 49.09 | 40.13 | 54.62 | 57.85 |
| Random (5K) | 11.34 | 23.86 | 36.53 | 42.83 | 46.93 | 0.21 | 5.73 | 4.16 | 23.36 | 34.97 | 21.17 | 36.44 | 46.65 | 49.85 | 48.03 | 38.88 | 53.32 | 56.85 |
| Random (1K) | 10.45 | 22.45 | 35.77 | 42.55 | 45.93 | 0.18 | 5.4 | 3.75 | 22.0 | 33.9 | 21.37 | 36.01 | 46.0 | 49.92 | 47.37 | 38.38 | 52.56 | 56.11 |
| Zero-Shot (Unsupervised) | 7.66 | 11.33 | 27.95 | 37.35 | 41.99 | 7.97 | 7.37 | 14.41 | 23.17 | 32.92 | 11.04 | 25.82 | 35.91 | 42.91 | 38.69 | 26.18 | 45.46 | 48.72 |

Table 18: Full ablation results with all our 18 mLLMs. Avg. BLI scores (P@1×100%) on 20 XLING BLI directions. Row 1 ∼ 2: five-shot prompting with in-context examples extracted from nearest neighbours (NN) in $\mathcal{D}_S$ of size 5K and 1K. Row 3 ∼ 4: five-shot prompting with random in-context examples in $\mathcal{D}_S$ of size 5K and 1K. Row 5: zero-shot prompting without any in-context examples.

| Ground-Truth Translation Pair | Method | In-Context Examples | Prediction |
|---|---|---|---|
| brisati (HR) → **erase** (EN) | 0-Shot | - | break |
| | 5-Shot | izbrisati→delete, kopirati→copy, ispraviti→amend, dopuniti→supplement, pisati→write | **erase** |
| jestiv (HR) → **edible** (EN) | 0-Shot | - | joyful |
| | 5-Shot | jesti→eat, češnjak→garlic, povrće→vegetable, grašak→peas, biljka→plant | **edible** |
| slap (HR) → **waterfall** (EN) | 0-Shot | - | slap |
| | 5-Shot | slapovi→falls, kanjon→canyon, jezero→lake, potok→creek, potoci→streams | **waterfall** |
| valcer (HR) → **waltz** (EN) | 0-Shot | - | valve |
| | 5-Shot | orkestar→orchestra, koncert→concert, balet→ballet, simfonija→symphony, klavir→piano | **waltz** |
| šaran (HR) → **carp** (EN) | 0-Shot | - | shark |
| | 5-Shot | pastrva→trout, riba→fish, ribolovac→fisher, gavran→raven, bizaran→bizarre | **carp** |
| sove (HR) → **owls** (EN) | 0-Shot | - | slept |
| | 5-Shot | sova→owl, životinje→animals, ptica→bird, mačke→cats, životinja→animal | **owls** |
| gušter (HR) → **lizard** (EN) | 0-Shot | - | gusher |
| | 5-Shot | štakor→rat, zmija→snake, zvijer→beast, kornjača→turtle, majmun→monkey | **lizard** |
| sezam (HR) → **sesame** (EN) | 0-Shot | - | cumin |
| | 5-Shot | grašak→peas, rajčica→tomato, povrće→vegetable, rajčice→tomatoes, smokva→fig | **sesame** |
| odgođena (HR) → **postponed** (EN) | 0-Shot | - | **postponed** |
| | 5-Shot | odgoditi→delay, bila→been, trebala→supposed, najave→announcements, provedena→conducted | delayed |
| tajnost (HR) → **secrecy** (EN) | 0-Shot | - | **secrecy** |
| | 5-Shot | krajnost→extreme, odanost→loyalty, odlučnost→resolve, tajna→secret, točnost→accuracy | discretion |
| prognoza (HR) → **prognosis** (EN) | 0-Shot | - | **prognosis** |
| | 5-Shot | prognoze→forecasts, dijagnoza→diagnosis, terapija→therapy, liječenje→treatment, astma→asthma | forecasts |
| otapanje (HR) → **dissolution** (EN) | 0-Shot | - | **dissolution** |
| | 5-Shot | doziranje→dosage, kiseline→acids, filtriranje→filtering, dobivanje→obtaining, curenje→leak | melting |
| sopportare (IT) → **endure** (EN) | 0-Shot | - | support |
| | 5-Shot | alleviare→ease, portare→bring, superare→exceed, odiare→hate, minimizzare→minimize | **endure** |
| snello (IT) → **slender** (EN) | 0-Shot | - | quick |
| | 5-Shot | flessibile→flexible, coda→tail, sottile→subtle, leggermente→slightly, aspetto→appearance | **slender** |
| hindi (IT) → **hindi** (EN) | 0-Shot | - | english |
| | 5-Shot | india→india, indiano→indian, tailandese→thai, lingua→tongue, arabo→arabic | **hindi** |
| velo (IT) → **veil** (EN) | 0-Shot | - | bicycle |
| | 5-Shot | cappello→hat, mantellina→cape, viso→face, cuscino→pillow, vestito→dress | **veil** |
| prugna (IT) → **plum** (EN) | 0-Shot | - | prickly |
| | 5-Shot | ciliegia→cherry, vaniglia→vanilla, zenzero→ginger, marmellata→jam, aglio→garlic | **plum** |
| propano (IT) → **propane** (EN) | 0-Shot | - | methane |
| | 5-Shot | idrogeno→hydrogen, acido→acid, carbonio→carbon, combustibili→fuels, polimero→polymer | **propane** |
| pomello (IT) → **knob** (EN) | 0-Shot | - | apple |
| | 5-Shot | maniglia→handle, regolabile→adjustable, maniglie→handles, coperchio→lid, cinghia→strap | **knob** |
| altopiano (IT) → **plateau** (EN) | 0-Shot | - | high |
| | 5-Shot | montagne→mountains, pianure→plains, vallata→dale, pianura→plain, valle→valley | **plateau** |
| eventuale (IT) → **eventual** (EN) | 0-Shot | - | **eventual** |
| | 5-Shot | necessario→necessary, valutare→gauge, possibile→possible, necessaria→needed, richiedere→require | possible |
| scopre (IT) → **discovers** (EN) | 0-Shot | - | **discovers** |
| | 5-Shot | scoprire→discover, crede→believes, chiede→asking, spiega→explains, ragazza→girl | discover |
| rane (IT) → **frogs** (EN) | 0-Shot | - | **frogs** |
| | 5-Shot | topi→mice, animali→animals, rana→frog, mosche→flies, creature→creatures | snakes |
| sorprese (IT) → **surprises** (EN) | 0-Shot | - | **surprises** |
| | 5-Shot | sorpresa→surprise, inaspettato→unexpected, sorprendente→surprising, aspettandosi→expecting, aspettative→expectations | surprised |

Table 19: Translation examples on HR→EN and IT→EN. We include here ground truth translation pairs and show the predictions derived from zero-shot and five-shot prompting with LLaMA₁₃ᵦ.