# OpenReview forum: "On Bilingual Lexicon Induction with Large Language Models"
_EMNLP/2023/Conference — EMNLP 2023 Main_

### Official Review · Reviewer_cKpL · 2023-07-30

**Soundness:** 4

**Excitement:**

4: Strong: This paper deepens the understanding of some phenomenon or lowers the barriers to an existing research direction.

**Paper Topic And Main Contributions:**

The paper focuses on Bilingual lexicon induction (BLI). The novelty relies on using multilingual LLMs and prompting to induce the translation pairs. The article includes a comprehensive set of main and side experiments. They obtain competitive performances, particularly using models like LLaMA




**Questions For The Authors:**

Could BLI be affected by the fact that LLMs tend to be biased toward dominant languages like English?

**Reasons To Accept:**

- I find the article well structured and easy to read. The experimental setting seems sound.

- The authors claim that this work is the first one to explore BLI with multilingual LLMs.


**Reasons To Reject:**

Some minor weaknesses

- The authors say they include a typologically diverse sample. However, the criterion for choosing the languages seems more opportunistic (whichever languages are available on the current datasets). This is not necessarily bad; I just recommend being more explicit about this. If we look at Appendix A, we can see that most languages are actually Indo-European. In fact, in this table (Appendix A), the authors write ”linguistic family” but they are referring to linguistic genus, so the table needs to be corrected.

- The results show bad performance in low-resource scenarios. In a way, this is something we already knew with previous approaches for BLI: bad quality embeddings lead to deficient induction of translation pairs.

- Large language models, e.g., LLaMA might be biased towards English.


**Reproducibility:**

4: Could mostly reproduce the results, but there may be some variation because of sample variance or minor variations in their interpretation of the protocol or method.

**Reviewer Confidence:**

3: Pretty sure, but there's a chance I missed something. Although I have a good feel for this area in general, I did not carefully check the paper's details, e.g., the math, experimental design, or novelty.

---

> ### Author Rebuttal · Authors · 2023-08-27
>
> Thanks for your positive feedback and helpful suggestions! Here we respond to the three minor weaknesses and the question for the authors respectively.
>
> **Minor Weakness 1.** Thanks for pointing this out! We admit that the existing BLI datasets actually do not cover a really large amount of diverse languages and we will be more explicit about it (we also already mentioned in L644-L658, that the range of languages for LLMs is one limitation of our method and LLMs for BLI in general). We agree that ‘language genus’ would be more proper and will adopt more accurate wording following your suggestion.
>
> **Minor Weakness 2.** We agree that previous approaches for BLI also show lower results in low-resource scenarios, and we understand that similar findings for LLMs may not make readers feel more excited. However, please consider the BLI performance in the following four setups: (A) traditional methods on high-resource language pairs (B) traditional methods on low-resource language pairs (C) LLMs on high-resource language pairs and (D) LLMs on low-resource language pairs. Previous work tells us that A>B, and analogously we can guess (before reading our paper) C>D. But some **more findings validated by our experiments** are that e.g., for LLaMA-13b we show C>A but D<B. How to further boost language coverage and capabilities of LLMs for lower-resource languages is an open question for the entire field of multilingual NLP, and some improvements on that front should also be reflected in BLI results as well in future research.
>
> **Minor Weakness 3.** We fully agree and acknowledge that LLMs are biased towards English. Our work shows that even such biased LLMs (e.g., LLaMA) can already outperform traditional BLI methods on many language pairs including non-English pairs. If stronger, more multilingual and less biased LLMs get developed in the future, our method may achieve even better results. Thank you for **implicitly** indicating that our method will potentially become more influential in the future, we will expand the discussion of (currently limited) multilinguality of LLMs (with the focus on the BLI task) following your feedback.
>
> **Question For The Authors.** Our preliminary conclusion is that LLMs also achieve strong BLI performance for non-English pairs. In our appendices, Tables 12,13,15 show detailed results on each language pair for the XLING dataset. For example, LLaMA supports 6 out of 8 languages. The 6 languages result in 30 language pairs and 20 of them are between non-English languages. Here are some statistics comparing LLaMA-13b with CLWE-based SotA:
>
> | |10 Language pairs involving English|20 non-English Language pairs |All 30 pairs|
> |---|---|---|---|
> |5K Setup|LLaMA wins in 8/10 cases|LLaMA wins in 16/20 cases|LLaMA wins in 24/30 cases|
> |1K Setup|LLaMA wins in 5/10 cases|LLaMA wins in 18/20 cases|LLaMA wins in 23/30 cases|
> |Zero-shot Setup|LLaMA wins in 5/10 cases|LLaMA wins in 7/20 cases|LLaMA wins in 12/30 cases|
>
> From this table we did **not** find very clear clues showing that LLaMA-13b may favour language pairs that involve English. We only know that LLMs (with <=13b parameters) may struggle with low-resource languages (corresponding to Minor Weakness 2 above).

---

### Official Review · Reviewer_om7d · 2023-08-02

**Soundness:** 5

**Excitement:**

4: Strong: This paper deepens the understanding of some phenomenon or lowers the barriers to an existing research direction.

**Paper Topic And Main Contributions:**

This work investigates large text-to-text language models for bilingual lexicon
induction, a task that was not explored with LLMs before. The authors compare
various models, including different model types (encoder-decoder and
decoder-only) and sizes, using various prompt templates, in zero- and few-shot
(in-context learning and model fine-tuning) setups. Although the main goal of
the paper is the analysis of LLMs on BLI, it contributes novel techniques
as well, such as in-context learning with nearest neighbors examples or prompt
templates for the BLI task. The experiments are performed on various language
pairs, including high-resource and moderate-resource pairs. On top of presenting
LLM's BLI performance in general, the authors perform various analyses and
ablation studies. The paper is detailed and easy to follow.


**Questions For The Authors:**

* A: Although P@1 shows similar trends as P@5 of MRR, I think it would be
interesting to explore top 5/10 LLM outputs. Static cross-lingual embeddings,
such as aligned fastText vectors often retrieve morphological variants of the
same word in the top-n output. Would LLMs behave similarly or maybe output a more
varied set of words, e.g., synonyms?
* B: BLI of named entities is often problematic with static word embeddings, e.g.,
the output contains a set of politicians of the given language/country instead of
the correct translation. Would LLMs be different?
* C: I am wondering whether the authors have an intuition of why LLaMA has the
best performance and whether this finding could correlate with the performance
on other cross-lingual tasks?
* D: I am also wondering about the long-term implications/use of the findings.
The authors mention a few applications of BLI, such as using it to generate
translation options or embeddings to initialize unsupervised MT systems, however
end-to-end MT is possible with LLMs and no embeddings were involved in this
work. Would BLI performance correlate with other cross-lingual task performance
for example?


**Reasons To Accept:**

* LLMs were not explored for BLI before.
* The paper presents detailed experiments and analyses, giving an extensive
overview of LLM's lexical capabilities.


**Reasons To Reject:**

* No significant reasons to reject.


**Reproducibility:**

5: Could easily reproduce the results.

**Reviewer Confidence:**

4: Quite sure. I tried to check the important points carefully. It's unlikely, though conceivable, that I missed something that should affect my ratings.

---

> ### Author Rebuttal · Authors · 2023-08-27
>
> Many thanks for your feedback and encouragement! Here we respond to the questions:
>
> **Question A.** P@1 is still the most standard and most used metric in BLI, hence it is adopted in our work. Concerning P@K and MRR, by manually inspecting the generated output texts, we found that many output words and phrases that rank 2nd, 3rd, 4th or 5th according to the beam score also make sense. Therefore, the trend for P@5/10, MRR should also hold (previous work, mentioned in footnote 16, states that the same trend is observed for P@1, P@5/10 or MRR).
>
> **Question B.** This is a very interesting observation. We are not familiar with the standard BLI-for-named-entities tasks and benchmarks, and this would definitely be a very interesting analysis. Following your suggestion, we will run a small-scale side analysis in the current work, but will leave the bulk of additional exploration for future research.  Concerning the issue that the output may contain “politicians of a given language instead of the correct translation”, our intuition is that maybe providing some in-context examples (few-shot learning) like what is done in our work could help, which can ‘teach’ LLMs to produce better output that is more likely to comply with some specific requirements (domain, content, format, etc.) as demonstrated in the in-context examples - this is another interesting experiment to try out.
>
> **Question C.** LLaMA is the 2023 new SotA LLM pretrained on a larger corpus compared to earlier LLMs. LLaMA also adopts a series of recent improvements such as (1) rotary embeddings, (2) SwiGLU activation, and (3) pre-normalization which are highlighted in the LLaMA paper. Please see our response to Question D below to see how BLI can improve NMT.
>
> **Question D.** Here we show how our text-to-text BLI (without calculating word embeddings) can potentially improve **end-to-end NMT**. Once BLI can improve NMT, the “Translate Train” and “Translate Test” **cross-lingual transfer learning** approaches can be improved accordingly which rely on the quality of NMT. Following your question and suggestion, we will expand the discussion on the relationship between BLI and NMT in the context of LLMs.
>
> - First, we found some initial recent works on using bilingual dictionaries to improve end-to-end NMT [1,2], where pure texts are used and no word embeddings are calculated at all. E.g., in order to translate a sentence $S_{x}$ in language X into language Y, and given some useful word translation pairs ($w_{x}^1$, $w_{y}^1$), ($w_{x}^2$, $w_{y}^2$) all in pure texts, [1] designs templates for prompting: “Translate the following sentence to Y: $S_{x}$. In this context, the word $w_{x}^1$ means $w_{y}^1$, the word $w_{x}^2$ means $w_{y}^2$. The full translation to Y is:”. [2] presents two related but different tricks.
>
>     - [1] Dictionary-based Phrase-level Prompting of Large Language Models for Machine Translation. Ghazvininejad et al. arXiv. 2023. (please see its Figure 1 as an example)
>     - [2] BILEX Rx: Lexical Data Augmentation for Massively Multilingual Machine Translation. Jones et al. arXiv.2023.
>
> - Second, previous work [1-2] actually uses ground-truth bilingual dictionaries to aid NMT. The BLI output can supplement gold-standard dictionaries in two scenarios: (1) BLI can provide word translations for words and language pairs not covered in existing gold-standard dictionaries; (2) BLI can provide multiple (top k) possible translations for downstream NMT models to take into account.

---

### Official Review · Reviewer_5V65 · 2023-08-16

**Typos Grammar Style And Presentation Improvements:** A
**Soundness:** 4

**Excitement:**

4: Strong: This paper deepens the understanding of some phenomenon or lowers the barriers to an existing research direction.

**Paper Topic And Main Contributions:**

This paper explores the use of large language models (LLMs) for the task of bilingual lexicon induction (BLI). It attempts to answer whether and to what extent it is possible to prompt and fine-tune LLMs for BLI and how do they compare on standard BLI benchmarks. In addition to zero-shot prompting of existing LLMs, the authors examine in-context few-shot prompting. This work demonstrates the strong performance of this approach for high-resource languages and in few shot approaches, when compared to previous approaches using cross-lingual word representations. However, when looking at the results for low-resource languages, the proposed approach with prompting out-of-the-box language models is still lagging behind previous methods. In addition, this paper shows that fine-tuning a smaller model is less successful than prompting larger models out-of-the-box, which could be a finding of interest for other tasks. Regarding the novelty of this paper, LLMs have already shown success in machine translation which makes it expected that they will perform well for the related task of BLI. Nevertheless, this paper presents an explicit evaluation on BLI benchmarks, which is a significant contribution.

**Questions For The Authors:**

A: The paper is lacking insights into the stability of the results (e.g. there is no mention of repeating experiments for different random seeds). The authors should add this information, especially in the context of prompting LLMs and the choice of few-shot examples. Furthermore, when reporting averages, standard deviations should be added, in addition to mean scores, e.g. when averaging multiple language pairs in Tables 1 and 3 , also potentially in Figures 1 and 2. This concern regarding the stability to different random initializations should also be addressed in the fine-tuning experiments (Tables 4 and 17).

**Reasons To Accept:**

The primary strengths of this paper are tied to its thorough examination of the topic. It encompasses 5 distinct LLM families, including both encoder-decoder and decoder-only models. It also covers 20 high-resource and 6 low-resource language pairs, from 2 standard BLI benchmark datasets, while comparing to strong baselines from previous work on this topic. Given the already acknowledged of power of LLMs, this work contributes a comprehensive evaluation for BLI tasks and contributes an overview of the capabilities and limitations to the research field.

**Reasons To Reject:**

The main concern is the lack of insights into the stability of the presented results. The paper doesn't state whether the experiments were repeated multiple times, and the degree to which randomness influences the outcomes when language models are prompted.

**Reproducibility:**

5: Could easily reproduce the results.

**Reviewer Confidence:**

4: Quite sure. I tried to check the important points carefully. It's unlikely, though conceivable, that I missed something that should affect my ratings.

---

> ### Author Rebuttal · Authors · 2023-08-27
>
> We thank the reviewer for their feedback, and hope that our clarifications below will remove the major concerns and motivate the reviewer to reassess their initial scores.
>
> >**Reasons To Reject - Concerns on the Stability of results / Randomness.** This is mentioned as the only “Reasons to Reject” and also asked in the “Questions for the Authors”.
>
> **Response: Our method (in the main experiments and without fine-tuning) expects to yield non-varied output (without any significant randomness)**, as also confirmed in our preliminary experiments, given the same/fixed prompt. In more detail:
>
> 1. For our **main experiments (the whole Tables 1-2, 12-16, Figures 1-2)** the models are fixed (downloaded from Huggingface), and are not fine-tuned at all. The choice of in-context examples is nearest neighbour (NN) retrieval in the fastText embedding space, as stated in L298-L308, which is a deterministic choice. For generation, we use deterministic beam search without any randomness (rather than beam sampling). **The whole process is completely deterministic, does not imply any stochasticity**, and should be fully reproducible as such.
>
> 2. We stated in footnote 7 that our few-shot “learning” also does not involve any fine-tuning. The term “few-shot learning” in our setup is inherited from the GPT-3 paper, where several in-context examples are appended to the input sentence but there is also no fine-tuning on downstream tasks.
>
> 3. For our main experiments, we provide extremely significant p-values (L530-L542), which support our claims.
>
> 4. The **randomness exists in two minor parts of our side analysis** as follows.
>      - The “Random 5K” and “Random 1K” rows in our ablation studies in Table 3 and also Table 17 (which is a more detailed version of Table 3). The ablation study also did not involve any fine-tuning. We did not run it multiple times because the results are significant enough to support our findings. After all, the results are averaged scores over 20 language pairs each including about 2k test samples (in total 38776 test samples). Here, we additionally estimate the significance level (p-values) related to “Random 5K” and “Random 1K” via $\chi^2$ test in the following Table to supplement our Table 3, and we’ll add this information into the appendix. Statistically significant: p<0.05 (also see our footnote 17). The results sufficiently prove the usefulness of in-context examples as stated in L567-L576.
> |   |  mT5-xxl |  mT0-xxl | mGPT  |  XGLM-4.5 | LLAMA-13b |
> |---|---|---|---|---|---|
> |P-value (NN-5k vs Random-5k)|  1.27e-60 |  1.95e-19 |  2.35e-19 | 4.67e-11  |  1.41e-21 |
> |P-value (NN-1k vs Random-1k)| 9.34e-22  | 9.13e-6  |  4.6e-4 |  6.4e-7 |  1.1e-6 |
> |P-value (Random5k vs Zero-shot)| 1.7e-43  |  1.8e-9 |  1.4e-83 | 1.0e-311  | 8.4e-114  |
> |P-value (Random1k vs zero-shot)| 2.4e-28  | 3.9e-3  |  3.2e-85 |  5.8e-289 | 2.8e-94  |
>
>     - Our fine-tuning experiments (Table 4) do imply some possible randomness during training. However, our analysis in L580-L588 already indicates that fine-tuning **smaller** models still obviously underperforms off-the-shelf **larger** models (this is our main claim/findings for the fine-tuning experiments as stated in L592-L598).
>
> 5. Because of our limited access to computational resources, we prioritized other more important experiments over multiple runs, but we will include a subset of results averaged across 3 or 5 runs in the camera-ready following your suggestion.
>
> > **Paper Topic And Main Contributions - Concerns on Novelty.**
>
> **Response:**
>
> - **We stress that we are the first to tackle BLI via LLMs** in a systematic and didactic fashion, aiming to understand the current capabilities of different models. We conduct extensive experiments and thorough analyses on BLI with LLMs and our proposed **retrieval-augmented few-shot learning** achieves strong results on many language pairs. Insightful results are presented through comparisons between LLMs, investigations on the influence of model size and the number of in-context examples, templates, BLI-oriented fine-tuning, etc. We also consider the finding that the LLMs provide sub-optimal performance for lower-resource languages as an important contribution and a reference building block for future research.
>
> - NMT and BLI are two very related but still distinct areas of research: previous research also showed how to combine BLI output to improve NMT, so gains in BLI might also translate to benefits in NMT research.
>
> > **Typos.**
>
> **Response:** Thanks for pointing this out. L1135-L1136 refers to the results on PanLex-BLI lower-resource languages. We will make it more clear in the next version.

---

### Meta-Review · Area_Chair_a7YY · 2023-09-19

**Recommendation:** 5

**Metareview:**

This paper examines the use of LLMs for the task of Bilingual Lexicon Induction (BLI), and in particular examine and evaluate different prompting strategies. They find that LLMs have strong capability to generate bilingual lexica.

Soundness: All reviewers agree that the soundness of the paper is very high (giving scores of 4/4/5), and that comprehensive experiments are performed to support the conclusions in the paper.

Excitement: Reviewers generally score high excitement for the paper (4/4/4) and list the readability and thoroughness as main reasons. Some challenges to excitement are raised in that (1) it is not surprising that LLMs can generate BLIs well, and (2) it is somewhat unclear why one would need an LLM-generated BLI instead of simply using the LLM (which generated the BLI) to perform a downstream task itself. The authors reply with an example in machine translation and point to 2 arXiv papers that investigate this issue.

---

### Decision · Program_Chairs · 2023-10-07

**Decision:**

Accept-Main

**Comment:**

This paper examines the use of LLMs for the task of Bilingual Lexicon Induction (BLI), and in particular examine and evaluate different prompting strategies. They find that LLMs have strong capability to generate bilingual lexica.

Soundness: All reviewers agree that the soundness of the paper is very high (giving scores of 4/4/5), and that comprehensive experiments are performed to support the conclusions in the paper.

Excitement: Reviewers generally score high excitement for the paper (4/4/4) and list the readability and thoroughness as main reasons. Some challenges to excitement are raised in that (1) it is not surprising that LLMs can generate BLIs well, and (2) it is somewhat unclear why one would need an LLM-generated BLI instead of simply using the LLM (which generated the BLI) to perform a downstream task itself. The authors reply with an example in machine translation and point to 2 arXiv papers that investigate this issue.